

**Ozone formation sensitivity study using machine learning**
**coupled with the reactivity of VOC species**
Junlei Zhan[1], Yongchun Liu[1*], Wei Ma[1], Xin Zhang[2], Xuezhong Wang[2], Fang Bi[2],
Yujie Zhang[2], Zhenhai Wu[2], Hong Li[2*]
1. Aerosol and Haze Laboratory, Advanced Innovation Center for Soft Matter Science
and Engineering, Beijing University of Chemical Technology, Beijing 100029, China
2. State Key Laboratory of Environmental Criteria and Risk Assessment, Chinese
Research Academy of Environmental Sciences, Beijing 100012, China
Correspondence: liuyc@buct.edu.cn; lihong@craes.org.cn





## Abstract

The formation of ground-level ozone ($O_3$) is dependent on both atmospheric chemical
processes and meteorological factors. Traditional models have difficulty assessing $O_3$
formation sensitivity in a timely manner due to the limitations of flexibility and
computational efficiency. In this study, a random forest (RF) model coupled with the
reactivity of volatile organic compound (VOC) species was used to investigate the $O_3$
formation sensitivity in Beijing from 2014 to 2016, and evaluate the relative importance
(RI) of chemical and meteorological factors to $O_3$ formation. The results showed that
the $O_3$ prediction performance using initial concentrations of VOC species ($R^2 = 0.87$)
was better than that using total VOCs (TVOCs) concentrations ($R^2 = 0.77$). Meanwhile,
the RIs of VOC species correlated well with their $O_3$ formation potentials (OFPs). $O_3$
formation presented a negative response to NOx, $PM_{2.5}$ and relative humidity, and a
positive response to temperature, solar radiation and VOCs. The $O_3$ isopleth curves
calculated by the RF model were generally comparable with those calculated by the
box model. $O_3$ formation shifted from a VOC-limited regime to a transition regime from
2014 to 2016. This study demonstrates that the RF model coupled with the initial
concentrations of VOC species could provide an accurate, flexible, and computationally
efficient approach for $O_3$ sensitivity analysis.



## 1. Introduction


Ground-level ozone ($O_3$) pollution, which can cause adverse human health effects
such as cardiovascular and respiratory diseases, has received increasing attention in
recent decades (Cohen et al., 2017). As important precursors of $O_3$, volatile organic
compounds (VOCs) in the atmosphere are oxidized to produce peroxyl radicals ($RO_2$)
and hydroperoxyl radicals ($HO_2$), which will accelerate the $NO$-$O_3$-$NO_2$ cycle, thus
leading to the accumulation of $O_3$ (Wang et al., 2017a). The production and loss of $RO_2$
and $HO_2$ are highly dependent on the concentration ratio of VOCs and NOx in the
atmosphere. Hence, atmospheric $O_3$ concentrations or production rates show a
nonlinear relationship with VOCs and NOx. Moreover, the $O_3$-VOC-NOx sensitivity is
readily influenced by VOC species (Tan et al., 2018), meteorological parameters (Liu
et al., 2020a; Liu & Wang 2020), and even atmospheric particulate matter (Li et al.,
2019), thus, exhibits high temporal and spatial variability. Therefore, it is urgent to
develop an accurate and highly efficient method for timely assessing the sensitivity
regime of $O_3$ production and evaluating the effectiveness of a potential measure on $O_3$
pollution control.
The sensitivity of $O_3$ formation can usually be analysed using observed indicators,
such as ozone production efficiency (OPE, $\Delta O_3/\Delta NOz$) (Wang et al., 2010; Lin et al.,
2011), $HCHO/NO_y$ (Martin et al., 2004), and $H_2O_2/NOz$ (or $H_2O_2/HNO_3$) (Sillman 1995;
Hammer et al., 2002; Wang et al., 2017a), observation-based model (OBM) (Vélez-
Pereira et al., 2021) and chemical transport models including community multiscale air



quality (CMAQ) (Djalalova et al., 2015) and Weather Research and Forecasting with
Chemistry (WRF-Chem) model (Wang et al., 2020a). The observed indicators can be
utilized to quickly diagnose the sensitivity regime of $O_3$ production. However, the
accuracy is sensitive to the precision of tracer measurements. In addition, this method
lacks the predictability of $O_3$ concentrations for policy-making. OBMs combine *in-situ*
field observations and chemical box models, which are built on widely-used chemistry
mechanisms (e.g., MCM, Carbon Bond, RACM or SAPRC), and applied to the
observed atmospheric conditions to simulate the *in-situ* $O_3$ production rate (Mo et al.,
2018). The sensitivity of $O_3$ production to various $O_3$ precursors, including NOx and
VOCs can be diagnosed based on the empirical kinetic modeling approach (EKMA) or
quantitatively assessed with the relative incremental reactivity (RIR). Chemical
transport models, which are driven by meteorological dynamics and incorporated with
the emissions of pollutants and the complex atmospheric chemical mechanism, provide
a powerful tool for simulating various atmospheric processes, including spatial
distribution, regional transport *vs.* local formation, source apportionment and
production rates of pollutants and so on (Sayeed et al., 2021). At present, OBMs are
widely used to investigate $O_3$ formation sensitivity in China. Previous studies indicated
that $O_3$ formation in urban areas of China is located in a VOC-limited or a transition
regime and varies with time and location (Ou et al., 2016; Wang et al., 2017a; Zhan et
al., 2021).
Although both OBMs and chemical transport models can assess the sensitivity of





$O_3$ production and predict the $O_3$ pollution level in a scenario of control measures, the
calculation accuracy is affected by the uncertainty of input parameters (Tang et al., 2011;
Yang et al., 2021b). In addition, both of them are time-consuming and expensive when
computational resources are considered. Thus, they are mostly applied to sampling
cases with a short time span (days or weeks) (Xue et al., 2014; Ou et al., 2016), and
identifying $O_3$ formation sensitivity in a timely manner is difficult. Compared to
traditional methods, machine learning (ML) is able to capture the main factors affecting
atmospheric $O_3$ formation in a timely manner with great flexibility (without the
constraints of time and space) and high computational efficiency (Wang et al., 2020c;
Grange et al., 2021; Yang et al., 2021a). Recently, ML based on convolutional neural
network (CNN), random forest (RF) and artificial neural network (ANN) models has
been applied in simulating atmospheric $O_3$ and shown good performance in $O_3$
prediction (Ma et al., 2020; Xing et al., 2020). For example, Ma et al. (Ma et al., 2021a)
simulated $O_3$ concentrations in the Beijing-Tianjin-Hebei (BTH) region from 2010-
2017 using an RF model that considered meteorological variables and output variables
from chemical transport models, and the correlation coefficient ($R^2$) between the
observed and modelled $O_3$ concentrations was greater than 0.8. Liu et al. (Liu et al.,
2021) also reported a high accuracy (80.4%) for classifying pollution levels of $O_3$ and
$PM_{2.5}$ at 1464 monitoring sites in China using an RF model. According to these previous
studies, the RF model has shown good performance in terms of prediction accuracy and
computational efficiency (Wang et al., 2016; Wang et al., 2017b).



However, many ML studies have used total VOCs (TVOCs) to simulate $O_3$
formation and rarely considered the effect of VOC species on $O_3$ formation sensitivity
(Feng et al., 2019; Liu et al., 2021; Ma et al., 2021a). Thus, they were unable to identify
the chemical reactivity of a single species to $O_3$ formation, which may lead to
underestimations or even misunderstandings of the role of VOCs in $O_3$ formation
because the same concentration of TVOCs with different compositions may lead to
different OPEs. In addition, VOCs react with OH radicals during atmospheric transport,
which is the most important sink of VOCs (Carlo et al., 2004; Liu et al., 2020b). Makar
et al. (Makar et al., 1999) reported that highly reactive species, such as isoprene, were
underestimated by 40% when the OH reactions were ignored. Other studies indicated
that the initial concentrations of VOCs, which account for the photochemical loss of
VOCs during transport, were more representative of pollution levels in the sampling
area than the observed VOCs (Yuan et al., 2013; Zhan et al., 2021). However, whether
the ML model can identify the connection between the reactivity of VOC species and
$O_3$ formation sensitivity has not been clarified.
In this study, we used the RF model to evaluate the prediction performance of
atmospheric $O_3$ using the TVOCs, measured VOC species and photochemical initial
concentration (PIC) of VOC species. We compared the relative importance (RI) of the
precursors (VOC species, NOx, $PM_{2.5}$, CO) and the meteorological parameters
(temperature, solar radiation, relative humidity, wind speed and direction) on $O_3$
formation in the summer of Beijing from 2014 to 2016. We also discussed the



possibility of connecting the RIs of VOCs with their OFPs and the changes in $O_3$-VOC-
NOx sensitivity based on the RF model from 2014 to 2016. Our study indicates that the
RF model combined with initial concentrations of VOC species can simulate $O_3$
concentrations well and provides a flexible and efficient tool for $O_3$ modelling in a near
real-time way.
**2. Methods**
**2.1 Sampling site and data**
The sampling site (40.04°N, 116.42°E) is located at the campus of Chinese
Research Academy of Environmental Sciences and was described in our previous work
(Zhang et al., 2021). Briefly, the station is located two kilometers from the north 4[th] ring
road and surrounded by a mixed residential and commercial area. The concentrations
of VOCs, NOx, CO, $O_3$ and $PM_{2.5}$ were measured at 8 m above ground level at this
location. Meteorological parameters, including temperature (T), relative humidity (RH),
wind speed and direction (WS&WD), solar radiation (SR), were monitored at 15 m
above ground level. VOCs were measured by an online commercial instrument (GC-
866, Chromatotec, France), which consisted of two independent analysers for detecting
C2-C6 and C6-C12 hydrocarbon components. More details about the observations can
be found in the Supplemental Materials (S1). The PICs of VOCs were calculated
according to the method reported in our previous work (Zhan et al., 2021) and the
Supplemental Materials (S2).
**2.2 Random forest model**
The random forest (RF) is a type of decision tree that can be used for classification



and regression (Breiman 2001). During the training process, the model creates a large
number of different decision trees with different sample sets at each node, and then
averages the scores of each decision tree as its final score to obtain more accurate results
that avoid large bias and overfitting (Breiman 2001). Approximately one-third of the
samples are excluded from the sample when the decision tree is built and used to
calculate the out-of-bag data error. Hence, RF can evaluate the RI of variables via out-
of-bag (OOB) data error (Svetnik et al., 2003),
$RI_i = \sum (errOOB2_i - errOOB1_i)/N$  (1)
where N represents the number of decision trees, and errOOB1 and OOB2 represent
the out-of-bag data error of feature $i$ before and after adding tiny data noise (Kohavi &
John 1997; Breiman 2001), respectively. The $RI_i$ reflects the response of the RF model
to feature $i$ after adding tiny data noise. It was used to evaluate the importance and
sensitivity of feature $i$ to $O_3$ formation in this study. More details about RI can be found
in the Supplemental Materials (S3). To verify the stability of the model, we interrupted
the continuity of the time series, fed the randomly arranged inputs to the model, and
performed a significance test on the RI. The results showed that there was no significant
difference among the different tests ($P > 0.05$, $R^2 > 0.97$).
**3. Results and discussion**
**3.1 Overview of air pollutants and meteorological conditions**

Figure 1 shows the time series of air pollutants and meteorological parameters

during the observations from 2014 to 2016. In 2014, 2015 and 2016, the wind direction

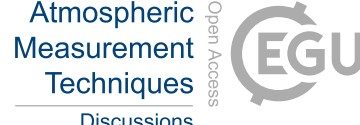

was dominated by northwest winds (Figure S1), with mean wind speeds of $3.1 \pm 2.7$ m
$s^{-1}$, $2.3 \pm 2.2$ m $s^{-1}$, and $1.3 \pm 1.2$ m $s^{-1}$, respectively, and the mean daytime temperature
were $22.3 \pm 5.8$, $23.9 \pm 5.0$ and $24.0 \pm 4.4$ °C, respectively. The average value of SR
decreased from 162.9 to 150.8 W $m^{-2}$ during the observation period. As shown in Figure
1F-G, in 2014, 2015 and 2016, the mean VOC concentrations were $20.3 \pm 10.9$, $15.8 \pm$
$8.3$ and $12.1 \pm 7.7$ ppbv, respectively, while the mean initial VOC concentrations were
$28.1 \pm 25.7$, $27.2 \pm 32.6$ and $16.4 \pm 16.1$ ppbv, respectively. Both the measured VOCs
and initial VOCs showed a decline along with a decrease in $PM_{2.5}$ concentration from
$67.2 \pm 53.5$ to $61.1 \pm 48.6$ μg $m^{-3}$ due to the Air Pollution Prevention and Control Action
Plan in China (Zhao et al., 2021). However, $O_3$ concentrations showed a slight upward
trend from $38.7 \pm 33.4$ to $42.7 \pm 27.9$ ppbv from 2014 to 2015 and then to $44.0 \pm 29.6$
ppbv in 2016. A similar trend was observed for NOx concentrations (Figure S2). As
shown in Figure 1F-G, the concentrations of four types (alkanes, alkenes, alkynes, and
aromatics) of VOCs showed significant differences from 2014 to 2016 due to the
variations in emission sources (Zhang et al., 2021). In addition to VOC species, the
variations in other parameters, such as meteorological conditions and $PM_{2.5}$, should
have a complex influence on $O_3$-VOC-NOx sensitivity (Li et al., 2019; Ma et al., 2021b).

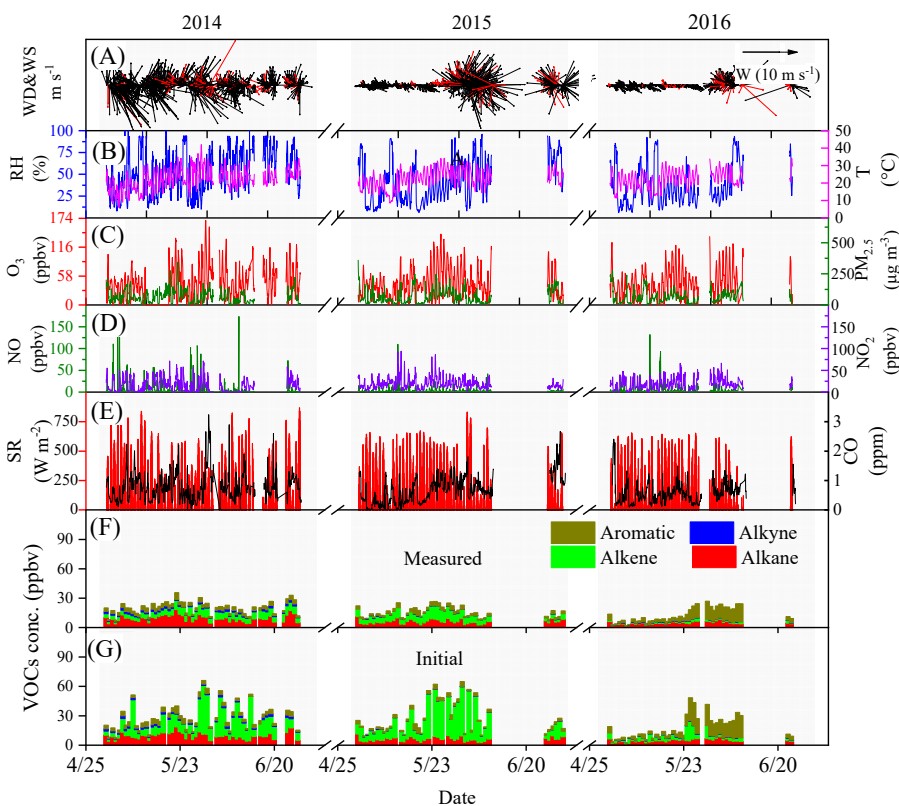


**Figure 1.** Time series of air pollutants and meteorological parameters during

observations in Beijing.

**3.2 Prediction performance of the model.**

To build a robust model, we evaluated the prediction performance of the RF model

for the ambient $O_3$ simulation. Figure 2 shows the $O_3$ prediction performance when

chemical species (including VOCs, NOx, $PM_{2.5}$, CO) and meteorological factors (i.e.,

WS, WD, SR, T and RH) were used as inputs in the RF model. The details of the

modelling and input parameters are shown in Table S1. Figure 2A-C shows the time

series of the measured and modelled $O_3$ concentrations, which were simulated using


the TVOCs, measured VOC species and initial VOC species as input variables along
with the same set of other parameters. The correlation coefficients ($R^2$) of the training
data were 0.88, 0.94 and 0.94 for the TVOCs, measured VOC species and initial VOC
species, respectively. The corresponding root mean squared errors (RMSEs) for the
predicted $O_3$ concentrations were 9.9, 9.3 and 9.1. Figure 2D-F shows the prediction
performance of the testing dataset under these three circumstances. When the TVOCs
were split into VOC species, the $R^2$ increased from 0.77 to 0.86 as the number of data
features increased. Therefore, the VOC composition has a significant influence on $O_3$
prediction using the RF model. Thus, our model has good prediction performance ($R^2$
= 0.87) when combined with the initial VOC species. In previous studies using TVOCs,
the influence of VOC composition was neglected (Liu et al., 2021; Ma et al., 2021a).
Therefore, our results indicate that the RF model can accurately predict $O_3$
concentrations when the concentrations of VOC species are considered and identify the
connection between the reactivity of VOC species and $O_3$ formation in the atmosphere.

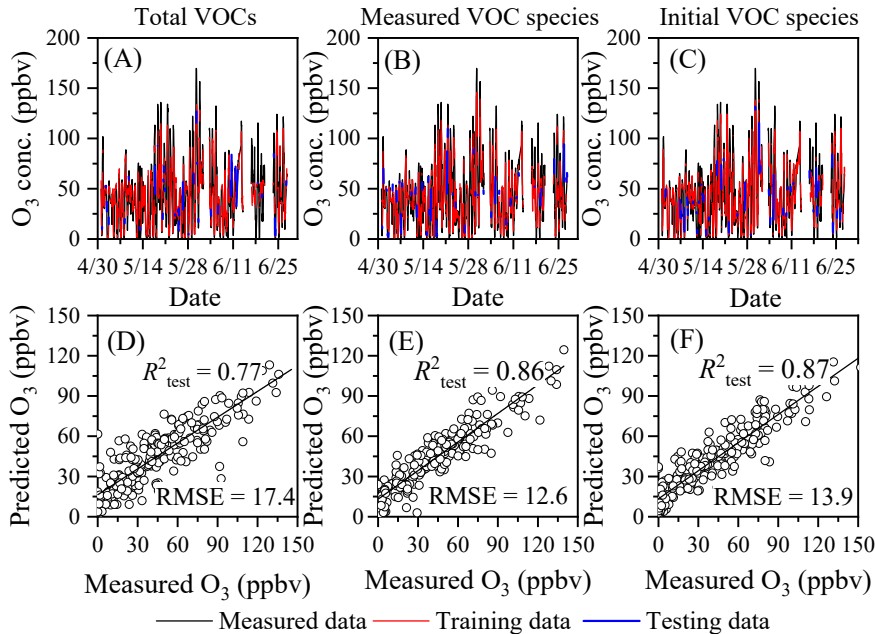


**Figure 2.** Comparison of the predicted and measured O₃ concentrations in Beijing in

the summer of 2014. (A and D: TVOC concentrations; B and E: measured

concentrations of VOC species; C and F: initial concentrations of VOC species)

**3.3 Relative importance of major factors**

Figure 3A shows the RIs of different ambient factors, including chemical and

meteorological variables on O₃ formation. The difference in the RIs is also compared

using the TVOCs and the VOC species as inputs. Chemical factors (including VOC

species, NOx, PM₂.₅ and CO) accounted for 83.1% of the contribution to O₃ production

in the summer of 2016. Meanwhile, VOC species accounted for approximately 66.7%

of O₃ production while the RIs using TVOC concentrations accounted for only 6.5%.

Ma et al. (Ma et al., 2021b) analysed the contribution of meteorological conditions and

chemical factors to O₃ formation on the North China Plain (NCP) using the CMAQ



model in combination with process analysis and found that chemical factors dominate
O₃ formation in summer. Using probability theory, Ueno et al. (Ueno & Tsunematsu
2019) also found that VOCs/NOx dominate O₃ production compared to meteorological
variables. Thus, our results are similar to those of previous studies based on chemical
models (Ueno & Tsunematsu 2019; Ma et al., 2021b), which demonstrates that the RF
model can reflect the contribution of VOC species to O₃ production even if the observed
VOC species are used.

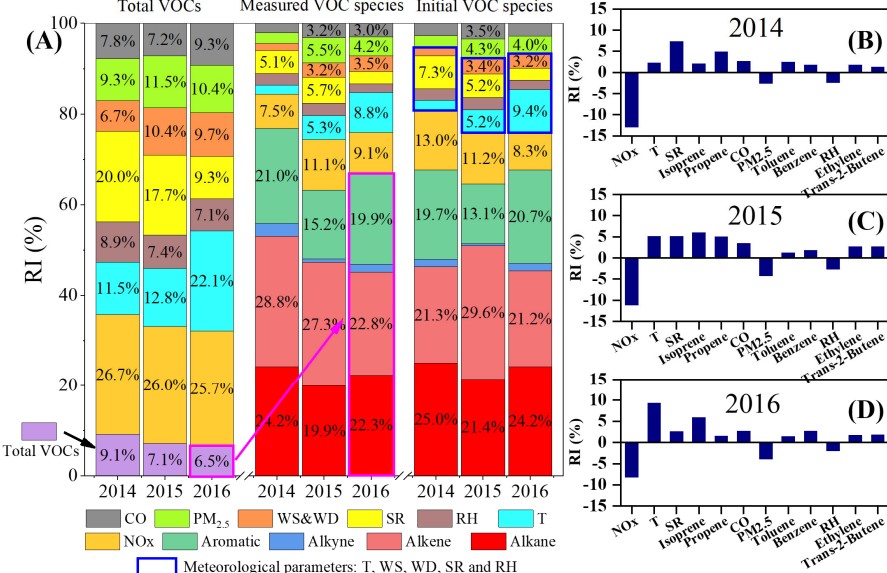


**Figure 3.** Percentage of RI for O₃ precursors and meteorological parameters (A) and
the top 12 factors with high values of RI in 2014-2016 (B-D: using initial concentrations
of VOC species).

Although ML is widely used to understand air pollution, explanations of ML

results (e.g., RI) are somewhat vague because ML is a black-box model (Sayeed et al.,
2021). Here, we compared the RIs of VOCs calculated using the initial VOC species



and the observed VOC species with the $O_3$ formation potentials (OFPs). The OFPs were
calculated by the maximum incremental reactivity (MIR) method (Carter 2010). As
shown in Figure S3, the RIs showed good correlations with the OFP. Interestingly, the
initial concentrations of VOC species improved the correlation coefficients between the
RIs and OFPs. Furthermore, we calculated the RIs and OFPs of different species using
the observed data during the campaign study in Daxing District in the summer of 2019
(Zhan et al., 2021), and a strong correlation was observed between the RIs of the initial
VOC species and the OFPs (Figure S4). These results indicate that the RIs of the initial
VOCs species in the ML model should partially reflect the chemical reactivity of VOCs
to produce $O_3$ in the atmosphere.

Although the RIs calculated using the initial VOC species slightly changed

compared to those calculated using the observed VOCs (Table S2), VOCs still
dominated $O_3$ formation (Figure 3A). For example, the initial VOCs dominated $O_3$
production in 2014, 2015, and 2016, with RI values of 67.7, 64.5 and 67.7%
respectively. Li et al. (Li et al., 2020a) used a multiple linear regression (MLR) model
to study the contribution of anthropogenic and meteorological factors to $O_3$ formation
in China from 2013-2019 and found that meteorological factors accounted for 36.8%
and anthropogenic factors accounted for 63.2%, which is similar to our results. Figure
3B-D shows the top 12 factors having a strongly influence on $O_3$ production.
Interestingly, NOx, $PM_{2.5}$ and RH showed negative responses to $O_3$ formation, while
other variables, including T, SR, CO and all of the VOCs, showed positive responses.

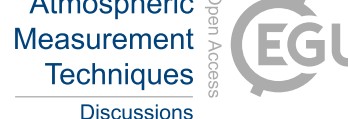

Thus, a decrease in NOx, PM$_{2.5}$ or RH will lead to an increase in O$_3$ concentration while
a decrease in T, SR, CO and VOCs will lead to a decrease in O$_3$ concentration. Although
O$_3$ formation is highly related to the photolysis of NO$_2$, a previous study demonstrated
that it is VOC-limited in summer in Beijing (Zhan et al., 2021). This finding is
consistent with the observed negative response of O$_3$ to NOx in this work. High
concentrations of PM$_{2.5}$ can reduce solar radiation and increase the sinks of reactive
radicals (HOx and ROx) (Li et al., 2019). In addition, high RH usually coincides with
low surface O$_3$ concentrations in field observations, which can be ascribed to the
inhibition of O$_3$ formation by the transfer of NO$_2$/ONO$_2$-containing products into the
particle phase and the promotion of dry deposition of O$_3$ on the surface (Kavassalis &
Murphy 2017; Yu 2019). These previous works can well explain the observed negative
response of O$_3$ to PM$_{2.5}$ and RH in Figure 3B. Previous studies have observed a positive
correlation between the O$_3$ concentration and T or SR (Steiner et al., 2010; Paraschiv
et al., 2020; Li et al., 2021). Temperature can directly affect the chemical reaction rate
of O$_3$ formation (Fu et al., 2015), and SR can promote the photolysis of NO$_2$ (Hu et al.,
2017; Wang et al., 2020b), thus accelerating O$_3$ formation. As mentioned above, O$_3$
formation is VOC-limited in Beijing; thus, a positive response of O$_3$ concentration to
VOCs is observed in Figure 3B. Interestingly, the RIs of isoprene showed an increasing
trend from 2014 to 2016 because of the obvious reduction in anthropogenic VOCs
(Figure 1) (Zhang et al., 2021). In the context of global warming, studies should focus
on the factors that affect O$_3$ formation, including biogenic emissions, T and SR. Thus,



additional efforts will be required to reduce anthropogenic pollutants in the future.
**3.4 Ozone formation sensitivity**
To further analyse the sensitivity of $O_3$ to VOCs and NOx from 2014 to 2016, we
plotted sensitivity curves for $O_3$ generation using the RF model, and the results are
shown in Figure 4A-C. Moreover, EKMA curves in 2015 were also obtained using the
OBM (Figure 4D). As shown in Figure 4A-C, $O_3$ formation was sensitive to VOCs in
the summer of Beijing during our observations, which is consistent with previous
studies that used box models (Li et al., 2020b) and chemical transport models (Shao et
al., 2021). This result is also consistent with the RIs of VOCs or NOx to $O_3$ formation
(Figure 3B). Interestingly, the $O_3$ formation sensitivity to VOCs decreases or gradually
shifts from the observed point to the transition regime from 2014 to 2016 (Figure 4A-
C), which is similar to that reported by Zhang et al. (Zhang et al., 2021). These
phenomena can be ascribed to the increased importance of meteorological factors, such
as T, SR, and RH, for $O_3$ formation and the variation in anthropogenic VOC emissions
(Steiner et al., 2010; Ma et al., 2021b).

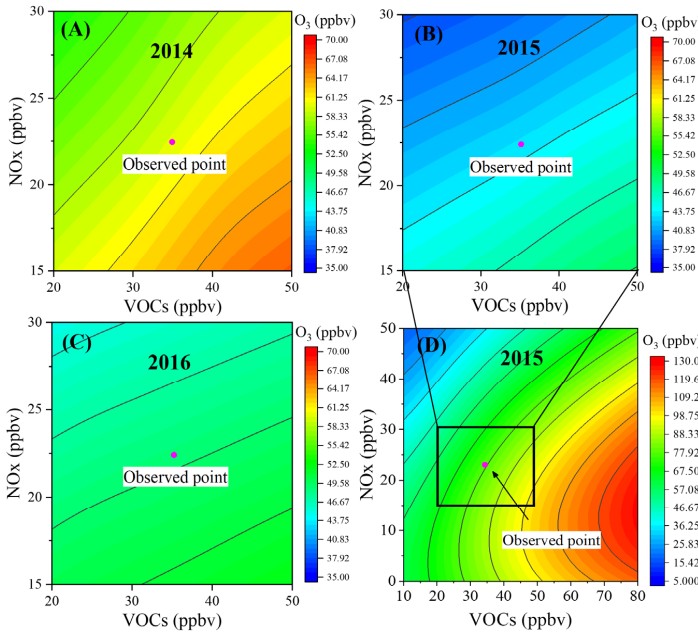

**Figure 4.** Ozone formation sensitivity curves from 2014-2016. (A, B, C: calculated by the RF model for 2014, 2015, and 2016, respectively. D: calculated by the OBM for 2015)

We compared $O_3$ sensitivity using the RF model based on the TVOCs and the initial VOC species in 2015. As shown in Figure S5, the $O_3$ concentrations predicted using the initial concentrations of VOC species were more accurate after correcting the reactivity during transport than those predicted using the TVOCs. Hence, a combination of the RF model and initial VOCs species (Figure 4B) can accurately depict the sensitivity regime of $O_3$ formation in comparison to the box model (Figure 4D), although a difference is observable between the predicted $O_3$ concentrations using these two models. In the box model, the $O_3$ isopleth plot was drawn with the maximum $O_3$ concentrations, while in the RF model, this plot was drawn with the real $O_3$



concentrations.
**4. Conclusions**
In summary, this work investigated $O_3$ formation sensitivity in the summer from
2014-2016 in Beijing using the RF model coupled with the reactivity of VOC species.
The results show that the prediction performance of $O_3$ by the RF model was
significantly improved when VOC species were considered compared to TVOCs.
Furthermore, after the photochemical loss of VOC species during transport was
corrected, the RIs of the VOC species were well correlated with the OFPs of VOC
species calculated using the MIR method, thus indicating that the RIs in the ML model
reflect the chemical reactivity of VOCs. Meanwhile, both NOx and highly reactive
species (such as isoprene, propene, benzene, and toluene) played an important role in
$O_3$ formation. An increased contribution of temperature to $O_3$ production was observed,
which implied the importance of temperature to $O_3$ pollution in the context of global
warming conditions. Both the RF model and the box model results showed that $O_3$
formation was sensitive to VOCs in Beijing, although the sensitivity regime shifted
from VOC-limited regime to a transition regime from 2014 to 2016. Due to the high
computational efficiency of ML, the $O_3$ formation sensitivity plotted by the RF model
coupled with the reactivity of VOC species can provide an accurate, flexible and
efficient approach for analysing $O_3$ sensitivity in a near real-time way.

**Code and data availability**



The code and datasets of VOCs and meteorology are available and will be provided by
the corresponding authors Yongchun Liu (liuyc@buct.edu.cn) and Hong Li
(lihong@craes.org.cn) upon request. The solar radiation data are publicly available via
www.copernicus.eu/en.
**Supplement**
Supplementary information is available for this paper.
**Author contributions**
Junlei Zhan designed the idea and wrote this manuscript; Yongchun Liu and Hong Li
provided useful advice and revised the manuscript; Wei Ma performed box model
simulations; and Xin Zhang, Xuezhong Wang, Fang Bi, Yujie Zhang and Zhenhai Wu
conducted the campaign and compiled the data. All authors contributed to the
discussion of the results and writing of the manuscript.
**Competing interest**
The authors declare that they have no conflict of interest.
**Acknowledgments**
This research was financially supported by the Ministry of Science and Technology of
the People's Republic of China (2019YFC0214701), the National Natural Science
Foundation of China (41877306 and 92044301) and the programs from Beijing
Municipal Science & Technology Commission (No. Z181100005418015). We thank
Yizhen Chen for providing the meteorological parameter data for campaign studies.



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
