# Peer review of "Ozone formation sensitivity study using machine learning"

_Atmospheric Measurement Techniques, 2021_

## Author Comment (AC1)

**1 Reviewer 1**

This is an interesting study, using machine learning to estimate the ozone formation sensitivity. The idea is not novel (a few previous studies with similar scope are cited in this manuscript). The method, using reactivity-corrected VOC measurements (i.e., initial VOC concentrations), sheds some insights into ozone production in an urban environment.

**Reply**: Thanks for your positive comments. We have carefully responded to all of your **point-by-point** comments and issues and have revised the manuscript accordingly.
These revisions are described in detail below.

10

11 However, there are several major issues:

Q1: (1) The machine learning workflow described in this manuscript does not include a robust or systematic solution to mitigate overtraining. I will elaborate on this later but the measures described in this work absolutely do not guarantee that overtraining is/can be avoided.

16 Reply: Thank you for your good suggestion. According to your suggestion, we performed a 12-fold cross-validation after data-normalization, i.e., by randomly 17 dividing the dataset into 12 subsets and alternately taking one subset as testing data and 18 the rest as training data. By doing this, every data point has an equal chance being 19 20 trained and tested. In lines 148-153 in the revised manuscript, we added a short paragraph "To avoid over-fitting, we trained the random forest model using cross-21 validation for the normalized data, which can improve the robustness of the model. 22 Briefly, we randomly divided the normalized data into 12 subsets, then alternately took 23 24 one subset as testing data along with the rest as training data. By doing this, every data 25 point has an equal chance being trained and tested.".

We added the RF model workflow to Text S3 in the revised Supporting Information.

28 "Text S3. Workflow of RF model and the calculation of Relative Importance (RI)
29 The workflow of RF model used in this study was established through the following
30 steps.

(1) Data description. The length of the input data from 2014 to 2016 were 1190, 1062
and 872 rows, respectively, in which different types of VOCs, NOx, CO, PM2.5 and
meteorological parameters (including temperature, relative humidity, solar radiation,
wind speed and direction) were used as input variables and O3 was the output variable.
The mean values (±standard deviation) of input/output parameters are shown in Table
S1.

(2) Data process. After the extreme values were removed, all data were normalized
(between 0 and 1) in order to decrease the sample distribution range, accelerate
calculation efficiency and improve the robustness of the RF model. Then, the dataset
was randomly divided into 12 subsets. Thus, a 12-fold cross-validation was performed
by alternately taking one subset as testing data and the rest as training data to ensure
that each data point has an equal chance being trained and tested.

(3) Hyper-parameters optimization. All network configuration parameters (i.e., leaf
number, number of trees, algorithm, and so on) were modified by a trial and error
method to obtain the optimized network structure. The optimized RF model parameters
are shown in Table S2. Figures S13 and S14 show the examples to optimize the number
of minimal samples split and trees, respectively.

48 (4) Model uncertainty estimation. The uncertainty of the model was estimated 49 according to the predicted and observed  $O_3$  concentrations. The performance of the 50 model was evaluated using R square ( $R^2$ ) and Root Mean Squared Error (RMSE).

(5) Relative importance (RI) calculation: The influence of an input variable on model performance was evaluated by changes in the accuracy of the model by variable permuting. Briefly, a change of prediction error was resulted from permuting a variable across the observations. The magnitude of the response was estimated using out-of-bag error of a predictor according to following steps.

For a random forest model that has T learners and p predictors in the training data, the first step is to identify the out-of-bag observations and the indices of the predictor variables that are split to a growing tree t (from 1 to T). Then, one can estimate the outof-bag error ( $\epsilon_t$ ) for each tree. For a predictor variable  $x_j$  (j: from 1 to p), one can estimate the model error ( $\epsilon_{t, j}$ ) again corroding to the out-of-bag observations after randomly 61 permuting the observations of xj. Thus, the difference of the model error  $(d_{t,j} = \varepsilon_{t,j} - \varepsilon_t)$ 62 is obtained. If the predictor variables are not split, the difference of a growing tree t is 63 0. The second step is to calculate the mean difference of the model errors  $(\overline{d}_j)$ , and the 64 standard deviation  $(\sigma_j)$  of the differences for all the learners and each predictor variable 65 in the training data. Finally, the out-of-bag relative importance (RI) for xj is calculated 66 by dividing the difference of the model errors by the standard deviation  $(\overline{d}_i/\sigma_i)$ .

(6) EKMA curves. The Empirical Kinetic Modeling Approach (EKMA) was used to 67 assess the O3 formation mechanism regime. Both the RF model and a box model with 68 Master Chemical Mechanism (MCM, 3.3.1) were used to calculate the EKMA curves. 69 For the RF model simulations, the observed point data was chosen as the mean values 70 71 of the input parameters during our observations, then the concentrations of VOCs and NOx were varied  $\pm 10\%$  (or from 90% to 110%) of their mean values with a step of 1% 72 in a two-dimensional matrix along with other inputs unchanged. This matrix was used 73 74 as the testing data, while all the measured data were taken as the training data in the RF 75 model to simulate O3 concentrations under different scenarios of VOCs and NOx concentrations. To decrease the model uncertainty, we set relatively small variations of 76 VOCs and NOx  $(\pm 10\%)$  compared to the observed values in this study. The mean 77 relative error of simulated O3 concentrations between RF model and Box model (within 78 15.6%, Figure S8) suggests that the RF model can well predict O3 concentrations during 79 our observations." 80

81

Q2: (2) Random forest depends heavily on the training dataset. The authors do not 82 83 provide an overview of the comprehensiveness of the training dataset: for instance, does the dataset cover all major chemical regimes in the EKMA plot, i.e., NOx-limited, 84 VOC-limited, NO titration? The authors claim that ozone production in Beijing, China 85 is mostly VOC-limited, which is consistent with previous studies. If the training set 86 collected in Beijing does not have sufficient coverage in the NOx-limited regime, then 87 the trained algorithm essentially attempts to extrapolate in that regime, which is 88 dangerous and prone to overtraining. I would then question the if this random forest 89

90 model can make meaningful forecast in that regime at all.

91 **Reply**: Thank you for your valuable suggestion. We added the description of training 92 dataset in Text S3 in the revised SI. This point has been replied in the aforementioned 93 question. The mean values (±standard deviation) of the input and output parameters for 94 the training data set are shown in Table R1. This Table was also added as Table S1 in 95 the revised SI.

**96**

**Table R1. An overview of training dataset from 2014 to 2016 during the observation**

97

**period.**

|                   | 2014            |       |             |      | 2015            |      |             |      | 2016            |      |        |       |
|-------------------|-----------------|-------|-------------|------|-----------------|------|-------------|------|-----------------|------|--------|-------|
| species / unit    | Measured
VOC |       | Initial VOC |      | Measured
VOC |      | Initial VOC |      | Measured
VOC |      | Initia | I VOC |
|                   | aver            | std.  | aver        | std. | aver            | std. | aver        | std. | aver            | std. | aver   | std.  |
|                   | age             | dev.* | age         | dev. | age             | dev. | age         | dev. | age             | dev. | age    | dev.  |
| Cyclopentane /    | 0.95            | 1.05  | 0.95        | 1.05 | 0.00            | 0.00 | 0.00        | 0.00 | 0.27            | 0.29 | 0.27   | 0.29  |
| ppbv              |                 |       |             |      |                 |      |             |      |                 |      |        |       |
| Ethane / ppbv     | 2.38            | 0.98  | 2.39        | 0.98 | 1.84            | 0.88 | 1.85        | 0.89 | 1.07            | 0.51 | 1.07   | 0.51  |
| Acetylene / ppbv  | 1.64            | 1.31  | 1.65        | 1.31 | 0.13            | 0.33 | 0.14        | 0.33 | 0.32            | 0.30 | 0.32   | 0.30  |
| Propane / ppbv    | 2.44            | 1.60  | 2.46        | 1.61 | 2.42            | 1.75 | 2.45        | 1.76 | 1.35            | 0.93 | 1.36   | 0.93  |
| Benzene / ppbv    | 0.60            | 0.44  | 0.61        | 0.44 | 0.47            | 0.35 | 0.47        | 0.36 | 4.59            | 4.23 | 4.64   | 4.29  |
| iso-Butane / ppbv | 0.95            | 0.66  | 0.96        | 0.67 | 0.35            | 0.53 | 0.35        | 0.54 | 0.24            | 0.18 | 0.24   | 0.19  |
| 2,2-              | 0.00            | 0.01  | 0.00        | 0.01 | 0.00            | 0.02 | 0.00        | 0.02 | 0.00            | 0.00 | 0.00   | 0.00  |
| Dimethylbutane /  |                 |       |             |      |                 |      |             |      |                 |      |        |       |
| ppbv              |                 |       |             |      |                 |      |             |      |                 |      |        |       |
| n-Butane / ppbv   | 1.57            | 1.11  | 1.60        | 1.11 | 0.67            | 0.87 | 0.69        | 0.89 | 0.85            | 0.73 | 0.87   | 0.74  |
| 2,2,4-            | 0.01            | 0.04  | 0.01        | 0.04 | 0.04            | 0.07 | 0.05        | 0.07 | 0.02            | 0.02 | 0.02   | 0.02  |
| Trimethylpentane  |                 |       |             |      |                 |      |             |      |                 |      |        |       |
| / ppbv            |                 |       |             |      |                 |      |             |      |                 |      |        |       |
| iso-Pentane /     | 0.11            | 0.38  | 0.11        | 0.40 | 0.00            | 0.00 | 0.00        | 0.00 | 0.16            | 0.18 | 0.16   | 0.18  |
| ppbv              |                 |       |             |      |                 |      |             |      |                 |      |        |       |
| 2,3-              | 0.07            | 0.08  | 0.07        | 0.08 | 0.06            | 0.08 | 0.06        | 0.08 | 0.02            | 0.03 | 0.02   | 0.03  |
| Dimethylpentane   |                 |       |             |      |                 |      |             |      |                 |      |        |       |
| / ppbv            |                 |       |             |      |                 |      |             |      |                 |      |        |       |
| 3-Methylhexane /  | 0.06            | 0.07  | 0.06        | 0.07 | 0.04            | 0.05 | 0.04        | 0.05 | 0.01            | 0.02 | 0.01   | 0.02  |
| ppbv              |                 |       |             |      |                 |      |             |      |                 |      |        |       |
| Toluene / ppbv    | 1.28            | 1.02  | 1.32        | 1.04 | 0.88            | 1.55 | 0.93        | 1.57 | 0.30            | 0.34 | 0.32   | 0.37  |
| 2,3-              | 0.00            | 0.00  | 0.00        | 0.00 | 0.00            | 0.03 | 0.00        | 0.03 | 0.06            | 0.08 | 0.06   | 0.08  |
| Dimethylbutane /  |                 |       |             |      |                 |      |             |      |                 |      |        |       |
| ppbv              |                 |       |             |      |                 |      |             |      |                 |      |        |       |
| n-Propyl benzene  | 0.01            | 0.02  | 0.01        | 0.02 | 0.01            | 0.03 | 0.01        | 0.03 | 0.04            | 0.11 | 0.05   | 0.11  |
| / ppbv            |                 |       |             |      |                 |      |             |      |                 |      |        |       |

| iso-Propyl         | 0.00 | 0.01 | 0.00 | 0.01 | 0.00 | 0.00 | 0.00 | 0.00 | 0.01 | 0.05 | 0.01 | 0.06 |
|--------------------|------|------|------|------|------|------|------|------|------|------|------|------|
| benzene / ppbv     |      |      |      |      |      |      |      |      |      |      |      |      |
| 2,3,4-             | 0.12 | 0.29 | 0.12 | 0.31 | 0.06 | 0.10 | 0.06 | 0.11 | 0.01 | 0.02 | 0.02 | 0.02 |
| trimethylpentane / |      |      |      |      |      |      |      |      |      |      |      |      |
| ppbv               |      |      |      |      |      |      |      |      |      |      |      |      |
| n-hexane / ppbv    | 0.37 | 0.30 | 0.39 | 0.31 | 0.05 | 0.18 | 0.06 | 0.20 | 0.18 | 0.27 | 0.19 | 0.30 |
| n-heptane / ppbv   | 0.08 | 0.09 | 0.09 | 0.10 | 0.06 | 0.06 | 0.06 | 0.07 | 0.02 | 0.02 | 0.02 | 0.02 |
| 2-methylhexane /   | 0.03 | 0.03 | 0.03 | 0.04 | 0.02 | 0.04 | 0.02 | 0.04 | 0.01 | 0.01 | 0.01 | 0.01 |
| ppbv               |      |      |      |      |      |      |      |      |      |      |      |      |
| 3-methylhexane /   | 0.01 | 0.02 | 0.01 | 0.02 | 0.01 | 0.02 | 0.01 | 0.02 | 0.00 | 0.01 | 0.00 | 0.01 |
| ppbv               |      |      |      |      |      |      |      |      |      |      |      |      |
| cyclohexane /      | 0.04 | 0.05 | 0.05 | 0.05 | 0.03 | 0.05 | 0.04 | 0.05 | 0.04 | 0.10 | 0.04 | 0.12 |
| ppbv               |      |      |      |      |      |      |      |      |      |      |      |      |
| ethylbenzene /     | 0.33 | 0.31 | 0.34 | 0.32 | 0.21 | 0.23 | 0.23 | 0.25 | 0.10 | 0.15 | 0.10 | 0.16 |
| ppbv               |      |      |      |      |      |      |      |      |      |      |      |      |
| n-octane / ppbv    | 0.04 | 0.11 | 0.04 | 0.11 | 0.00 | 0.00 | 0.00 | 0.00 | 0.00 | 0.00 | 0.00 | 0.00 |
| ethene / ppbv      | 2.15 | 1.36 | 2.31 | 1.43 | 1.72 | 1.16 | 1.90 | 1.25 | 0.39 | 0.30 | 0.41 | 0.31 |
| methylcyclohexa    | 0.01 | 0.03 | 0.01 | 0.03 | 0.01 | 0.03 | 0.01 | 0.04 | 0.02 | 0.03 | 0.02 | 0.04 |
| ne / ppbv          |      |      |      |      |      |      |      |      |      |      |      |      |
| n-nonane / ppbv    | 0.03 | 0.04 | 0.03 | 0.04 | 0.02 | 0.02 | 0.02 | 0.03 | 0.02 | 0.04 | 0.02 | 0.04 |
| n-decane / ppbv    | 0.02 | 0.04 | 0.02 | 0.05 | 0.02 | 0.03 | 0.02 | 0.03 | 0.00 | 0.01 | 0.00 | 0.01 |
| p-ethyltoluene /   | 0.06 | 0.08 | 0.06 | 0.08 | 0.02 | 0.03 | 0.03 | 0.04 | 0.07 | 0.10 | 0.07 | 0.11 |
| ppbv               |      |      |      |      |      |      |      |      |      |      |      |      |
| p-diethyl benzene  | 0.01 | 0.04 | 0.01 | 0.04 | 0.01 | 0.02 | 0.01 | 0.02 | 0.09 | 0.17 | 0.11 | 0.22 |
| / ppbv             |      |      |      |      |      |      |      |      |      |      |      |      |
| o-ethyl toluene /  | 0.03 | 0.04 | 0.04 | 0.04 | 0.01 | 0.03 | 0.01 | 0.03 | 0.08 | 0.28 | 0.09 | 0.32 |
| ppbv               |      |      |      |      |      |      |      |      |      |      |      |      |
| o-xylene / ppbv    | 0.09 | 0.18 | 0.10 | 0.18 | 0.16 | 0.18 | 0.19 | 0.20 | 0.14 | 0.26 | 0.15 | 0.27 |
| m-ethyl toluene /  | 0.02 | 0.07 | 0.02 | 0.07 | 0.04 | 0.09 | 0.04 | 0.09 | 0.03 | 0.04 | 0.03 | 0.05 |
| ppbv               |      |      |      |      |      |      |      |      |      |      |      |      |
| m-diethyl          | 0.01 | 0.03 | 0.01 | 0.03 | 0.00 | 0.01 | 0.00 | 0.01 | 0.00 | 0.02 | 0.00 | 0.02 |
| benzene / ppbv     |      |      |      |      |      |      |      |      |      |      |      |      |
| m/p-Xylene /       | 0.61 | 0.64 | 0.68 | 0.65 | 0.45 | 0.51 | 0.54 | 0.59 | 0.22 | 0.38 | 0.25 | 0.41 |
| ppbv               |      |      |      |      |      |      |      |      |      |      |      |      |
| propene / ppbv     | 2.07 | 1.18 | 2.83 | 2.26 | 4.40 | 2.61 | 6.60 | 6.12 | 0.28 | 0.41 | 0.34 | 0.45 |
| 1-Butene / ppbv    | 0.10 | 0.14 | 0.13 | 0.17 | 0.04 | 0.10 | 0.08 | 0.25 | 0.03 | 0.03 | 0.04 | 0.06 |
| 1-Pentene / ppbv   | 0.03 | 0.09 | 0.04 | 0.09 | 0.03 | 0.07 | 0.05 | 0.12 | 0.02 | 0.06 | 0.02 | 0.07 |
| 1,2,4-trimethyl    | 0.01 | 0.08 | 0.01 | 0.08 | 0.08 | 0.09 | 0.11 | 0.12 | 0.05 | 0.05 | 0.06 | 0.09 |
| benzene/ ppbv      |      |      |      |      |      |      |      |      |      |      |      |      |
| 1,2,3-trimethyl    | 0.00 | 0.01 | 0.00 | 0.01 | 0.03 | 0.05 | 0.04 | 0.08 | 0.05 | 0.28 | 0.05 | 0.28 |
| benzene/ ppbv      |      |      |      |      |      |      |      |      |      |      |      |      |
| a-pinene / ppbv    | 0.01 | 0.03 | 0.02 | 0.03 | 0.01 | 0.03 | 0.01 | 0.03 | 0.18 | 0.46 | 0.84 | 3.48 |
| cis-2-Butene /     | 0.34 | 0.70 | 0.85 | 2.67 | 0.66 | 0.85 | 1.77 | 4.56 | 0.04 | 0.05 | 0.11 | 0.29 |
| ppbv               |      |      |      |      |      |      |      |      |      |      |      |      |

| 1,3,5-                                | 0.05 | 0.07  | 0.08 | 0.11 | 0.03 | 0.05 | 0.07 | 0.14 | 0.25 | 0.56 | 1.07 | 4.11 |
|---------------------------------------|------|-------|------|------|------|------|------|------|------|------|------|------|
| Trimethylbenzene                      |      |       |      |      |      |      |      |      |      |      |      |      |
| / ppbv                                |      |       |      |      |      |      |      |      |      |      |      |      |
| styrene / ppbv                        | 0.18 | 0.27  | 0.30 | 0.61 | 0.00 | 0.03 | 0.01 | 0.08 | 0.27 | 0.79 | 0.57 | 2.08 |
| 2-methyl-1-                           | 0.18 | 0.37  | 0.72 | 2.94 | 0.04 | 0.04 | 0.26 | 1.68 | 0.02 | 0.09 | 0.03 | 0.12 |
| pentene / ppbv                        |      |       |      |      |      |      |      |      |      |      |      |      |
| trans-2-Butene /                      | 0.08 | 0.16  | 0.24 | 1.15 | 0.09 | 0.11 | 0.34 | 0.74 | 0.02 | 0.02 | 0.04 | 0.08 |
| ppbv                                  |      |       |      |      |      |      |      |      |      |      |      |      |
| cis-2-Pentene /                       | 0.15 | 0.20  | 0.37 | 0.93 | 0.17 | 0.17 | 0.91 | 4.24 | 0.01 | 0.02 | 0.02 | 0.08 |
| ppbv                                  |      |       |      |      |      |      |      |      |      |      |      |      |
| 1,3-Butadiene /                       | 0.09 | 0.10  | 0.19 | 0.34 | 0.04 | 0.05 | 0.12 | 0.38 | 0.02 | 0.03 | 0.05 | 0.25 |
| ppbv                                  |      |       |      |      |      |      |      |      |      |      |      |      |
| trans-2-Pentene /                     | 0.03 | 0.08  | 0.06 | 0.27 | 0.01 | 0.02 | 0.11 | 0.89 | 0.01 | 0.02 | 0.01 | 0.05 |
| ppbv                                  |      |       |      |      |      |      |      |      |      |      |      |      |
| $\beta$ -pinene / ppbv                | 0.00 | 0.01  | 0.01 | 0.03 | 0.01 | 0.01 | 0.02 | 0.15 | 0.00 | 0.01 | 0.00 | 0.02 |
| isoprene / ppbv                       | 0.89 | 0.64  | 5.70 | 18.7 | 0.34 | 0.43 | 6.40 | 21.5 | 0.13 | 0.17 | 2.12 | 7.46 |
|                                       |      |       |      | 8    |      |      |      | 6    |      |      |      |      |
| NO / ppbv                             | 7.03 | 17.02 | 7.03 | 17.0 | 3.38 | 5.59 | 3.38 | 5.59 | 5.28 | 10.3 | 5.28 | 10.3 |
|                                       |      |       |      | 2    |      |      |      |      |      | 5    |      | 5    |
| NO 2 / ppbv                | 15.5 | 15.79 | 15.5 | 15.7 | 19.1 | 12.6 | 19.1 | 12.6 | 18.7 | 12.4 | 18.7 | 12.4 |
|                                       | 0    |       | 0    | 9    | 1    | 8    | 1    | 8    | 2    | 0    | 2    | 0    |
| T/°C                                  | 22.5 | 6.28  | 22.5 | 6.28 | 22.7 | 5.24 | 22.7 | 5.24 | 22.3 | 4.85 | 22.3 | 4.85 |
|                                       | 6    |       | 6    |      | 0    |      | 0    |      | 7    |      | 7    |      |
| RH / %                                | 50.9 | 23.88 | 50.9 | 23.8 | 41.4 | 23.2 | 41.4 | 23.2 | 36.2 | 21.5 | 36.2 | 21.5 |
|                                       | 3    |       | 3    | 8    | 9    | 3    | 9    | 3    | 3    | 8    | 3    | 8    |
| SR / W m -2                | 162. | 222.9 | 162. | 222. | 153. | 205. | 153. | 205. | 150. | 199. | 150. | 199. |
|                                       | 92   | 5     | 92   | 95   | 29   | 01   | 29   | 01   | 81   | 35   | 81   | 35   |
| WS / m s -1                | 3.11 | 2.70  | 3.11 | 2.70 | 2.29 | 2.15 | 2.29 | 2.15 | 1.25 | 1.24 | 1.25 | 1.24 |
| WD / °                                | 162. | 105.0 | 162. | 105. | 175. | 101. | 175. | 101. | 184. | 108. | 184. | 108. |
|                                       | 42   | 7     | 42   | 07   | 38   | 87   | 38   | 87   | 21   | 06   | 21   | 06   |
| PM 2.5 /µg m -3 | 67.1 | 53.47 | 67.1 | 53.4 | 63.1 | 56.4 | 63.1 | 56.4 | 61.0 | 48.6 | 61.0 | 48.6 |
|                                       | 6    |       | 6    | 7    | 3    | 6    | 3    | 6    | 5    | 4    | 5    | 4    |
| CO /mg m -3                | 0.78 | 0.49  | 0.78 | 0.49 | 0.68 | 0.44 | 0.68 | 0.44 | 0.57 | 0.36 | 0.57 | 0.36 |
| O 3 / ppbv                 | 44.3 | 32.38 | 44.3 | 32.3 | 42.7 | 27.9 | 42.7 | 27.9 | 44.0 | 29.6 | 44.0 | 29.6 |
|                                       | 2    |       | 2    | 8    | 4    | 4    | 4    | 4    | 1    | 4    | 1    | 4    |

98 \* Standard Deviation (std. Dev.)

99

In Text S3 in the revised SI, we added a short paragraph "Data description. The length of the input data from 2014 to 2016 were 1190, 1062 and 872 rows, respectively, in which different types of VOCs, NOx, CO, PM2.5 and meteorological parameters (including temperature, relative humidity, solar radiation, wind speed and direction) 104 were used as input variables and O3 was the output variable. The mean values
105 (±standard deviation) of input/output parameters are shown in Table S1"

As shown in Figure R1 or Figure S15, the training dataset were located in VOC-106 limited, NOx-limited, and transition regimes, while most of the training data were 107 located in the VOC-limited regime. To avoid overtraining, we performed a 12-fold 108 cross-validation, i.e., by randomly dividing the observed data into 12 subsets and 109 alternately taking one subset as testing data and the rest as training data, to ensure that 110 111 each data point has an equal chance of being trained and tested. Figure R2 (Figure 2 in the revised manuscript) shows the comparisons between the measured and predicted O3 112 concentrations using different VOC inputs. The curves of the predicted O3 113 concentrations were spliced using the testing datasets in all runs. Thus, both the training 114 data and the testing data actually covered all the sensitivity regimes of O3 formation. 115 We think that the model is robust in the revised version according to your good 116 suggestion. 117

119 **Figure R1**. Sensitivity curves of O3 formation and distribution of training data in 2015.

---

## Author Comment (AC2)

**Reviewer 2#**

This article estimates the sensitivity of ozone formation using a random forest model with not only total VOCs concentrations but also observed concentrations and initial concentrations of VOC species. The result showed that the ozone prediction performance using initial concentrations of VOC species was better than that using total VOCs concentrations. Analytical reports using machine learning with total VOCs concentrations have been published recently. This article simply indicates the superiority of using overserved or initial concentrations of VOC species. From this aspect, the significance of this study is evident. The reviewer would recommend it for publication.

**Reply**: Thank you for your positive comments. We have carefully responded to all of your **point-by-point** comments and issues and have revised the manuscript accordingly. These revisions are described in detail below.

**Q1:** However, the reliability of analytical data and explanation of the initial concentration of VOC species are not sufficiently indicated in this article. And some expressions seem to be somewhat inadequate for well understanding. Slight revisions are required.

**Reply**: Thank you for your good suggestions. The reliability of analytical data will be replied in the following question. We have added more details about the initial concentration of VOC species. 1) We added the selection rules of tracers to calculate OH exposure and confirmed these rules based on our observation data. As shown in Figure R1 (Figure S9), the concentrations of xylene and ethylbenzene are well correlated, which indicates that they are simultaneously emitted. In addition, we compared the photochemical initial concentrations (PICs) calculated using xylene/ethylbenzene with that using toluene/benzene (Figure R2 or Figure S10). 2) We performed sensitivity tests about OH exposure calculation. The results showed that the uncertainty caused by the OH exposure (from −10% to +10%) ranged from 0.55 to 1.57 (Table R1 or Table S4). 3) we compared the chemical ages in this work with those reported in literatures. For example, the photochemical ages of isoprene were 0.01–6.21

h (1.26 ± 1.12 h). This value is comparable with previously reported photochemical ages (Shao et al., 2011; Gao et al., 2018). 4) The diurnal curves of measured and initial VOC concentrations were added in the revised SI (Figure R3 or Figure S12).

[Figure]

**Figure R1.** The relationship between xylene and ethylbenzene.

[Figure]

**Figure R2.** Comparison of the initial VOCs calculated using the ratio of xylene/ethylbenzene with that using the ratio of toluene/benzene in 2015. (Error bars are standard deviations.)

[Figure]

**Figure R3.** The daily variation of VOCs concentration. (A and D for 2014; B and E for 2015; C and F for 2016)

**Table R1**. $k_{OH}$, Method Detection Limit (MDL) and sensitivity test on estimation of $[OH]\times t$ of different VOC species

| number | species name | $k_{OH}$* | MDL** | Ratio to the initial VOC*** | | | | | |
| --- | --- | --- | --- | --- | --- | --- | --- | --- | --- |
| | | | | 2014 | | 2015 | | 2016 | |
| | | | | -10% [OH] ×t | +10% [OH] ×t | -10% [OH] ×t | +10% [OH] ×t | -10% [OH] ×t | +10% [OH] ×t |
| 1 | Ethane | 0.254 | 0.050 | 1.00 | 1.00 | 1.00 | 1.00 | 1.00 | 1.00 |
| 2 | Acetylene | 0.756 | 0.022 | 1.00 | 1.00 | 1.00 | 1.00 | 1.00 | 1.00 |
| 3 | Propane | 1.11 | 0.013 | 1.00 | 1.00 | 1.00 | 1.00 | 1.00 | 1.00 |
| 4 | Benzene | 1.22 | 0.011 | 1.00 | 1.00 | 1.00 | 1.00 | 1.00 | 1.00 |
| 5 | iso-Butane | 2.14 | 0.010 | 1.00 | 1.00 | 1.00 | 1.00 | 1.00 | 1.00 |
| 6 | 2,2-Dimethylbutane | 2.27 | 0.005 | 1.00 | 1.00 | 1.00 | 1.00 | 1.00 | 1.00 |
| 7 | n-Butane | 2.38 | 0.011 | 1.00 | 1.00 | 1.00 | 1.00 | 1.00 | 1.00 |
| 8 | 2,2,4-Trimethylpentane | 3.38 | 0.008 | 1.00 | 1.00 | 1.00 | 1.00 | 1.00 | 1.00 |
| 9 | iso-Pentane | 3.6 | 0.008 | 1.00 | 1.00 | 1.00 | 1.00 | 1.00 | 1.00 |
| 10 | Cyclopentane | 5.02 | 0.005 | 1.00 | 1.00 | 1.00 | 1.00 | 1.00 | 1.00 |
| 11 | n-hexane | 5.25 | 0.011 | 0.99 | 1.01 | 0.99 | 1.01 | 0.99 | 1.01 |
| 12 | Toluene | 5.58 | 0.009 | 1.00 | 1.00 | 0.99 | 1.01 | 1.00 | 1.00 |
| 13 | 2,3-Dimethylbutane | 5.79 | 0.004 | 1.00 | 1.00 | 1.00 | 1.00 | 0.99 | 1.01 |

| | | | | | | | | | |
|---|---|---|---|---|---|---|---|---|---|
| 14 | n-Propyl benzene | 5.8 | 0.008 | 1.00 | 1.00 | 1.00 | 1.00 | 0.99 | 1.01 |
| 15 | iso-Propyl benzene | 6.3 | 0.007 | 1.01 | 1.01 | 0.99 | 1.01 | 0.97 | 1.03 |
| 16 | 2,3,4-trimethylpentane | 6.6 | 0.008 | 0.99 | 1.01 | 0.99 | 1.01 | 1.00 | 1.00 |
| 17 | n-heptane | 6.81 | 0.009 | 0.99 | 1.01 | 0.99 | 1.01 | 0.99 | 1.01 |
| 18 | ethylbenzene | 7 | 0.009 | 0.99 | 1.01 | 0.99 | 1.01 | 0.99 | 1.01 |
| 19 | cyclohexane | 7.02 | 0.011 | 1.00 | 1.00 | 0.99 | 1.01 | 0.99 | 1.01 |
| 20 | 2,3-Dimethylpentane | 7.15 | 0.009 | 1.00 | 1.00 | 1.00 | 1.00 | 1.00 | 1.00 |
| 21 | 3-Methylhexane | 7.17 | 0.009 | 1.00 | 1.00 | 0.99 | 1.01 | 1.00 | 1.00 |
| 22 | ethene | 8.15 | 0.021 | 0.99 | 1.01 | 0.99 | 1.01 | 0.99 | 1.01 |
| 23 | n-octane | 8.16 | 0.008 | 0.99 | 1.01 | 1.00 | 1.00 | 1.00 | 1.00 |
| 24 | 2-Methylheptane | 8.31 | 0.008 | 1.00 | 1.00 | 0.99 | 1.01 | 0.99 | 1.01 |
| 25 | 3-Methylheptane | 8.59 | 0.008 | 1.00 | 1.00 | 1.00 | 1.01 | 0.99 | 1.01 |
| 26 | methylcyclohexane | 9.64 | 0.005 | 0.99 | 1.01 | 0.99 | 1.01 | 0.99 | 1.01 |
| 27 | n-nonane | 9.75 | 0.006 | 0.99 | 1.01 | 0.99 | 1.01 | 0.98 | 1.02 |
| 28 | n-decane | 11 | 0.007 | 0.99 | 1.01 | 0.99 | 1.01 | 0.99 | 1.01 |
| 29 | p-ethyl toluene | 11.8 | 0.007 | 0.99 | 1.01 | 0.98 | 1.02 | 0.98 | 1.02 |
| 30 | p-diethyl benzene | - | 0.008 | 1.00 | 1.00 | 0.99 | 1.01 | 0.97 | 1.03 |
| 31 | o-ethyl toluene | 11.9 | 0.007 | 0.99 | 1.01 | 0.99 | 1.01 | 1.00 | 1.00 |
| 32 | o-xylene | 13.6 | 0.007 | 0.99 | 1.01 | 0.98 | 1.02 | 1.00 | 1.00 |
| 33 | m-ethyl toluene | 18.6 | 0.010 | 0.99 | 1.01 | 0.99 | 1.01 | 0.97 | 1.03 |
| 34 | m-diethyl benzene | - | 0.009 | 0.99 | 1.01 | 0.99 | 1.01 | 0.98 | 1.02 |
| 35 | m/p-Xylene | 23.1/14.2 | 0.008 | 0.99 | 1.01 | 0.98 | 1.02 | 0.98 | 1.03 |
| 36 | propene | 26 | 0.015 | 0.96 | 1.04 | 0.95 | 1.05 | 0.96 | 1.05 |
| 37 | 1-Butene | 31.1 | 0.010 | 0.97 | 1.04 | 0.90 | 1.12 | 0.92 | 1.10 |
| 38 | 1-Pentene | 31.4 | 0.009 | 0.98 | 1.02 | 0.93 | 1.09 | 0.93 | 1.08 |
| 39 | 1,2,4-trimethyl benzene | 32.5 | 0.008 | 1.00 | 1.01 | 0.95 | 1.05 | 0.91 | 1.10 |
| 40 | 1,2,3-trimethyl benzene | 32.7 | 0.009 | 0.96 | 1.04 | 0.96 | 1.04 | 0.97 | 1.03 |
| 41 | a-pinene | 51.8 | 0.010 | 0.97 | 1.04 | 0.96 | 1.05 | 0.75 | 1.35 |
| 42 | cis-2-Butene | 55.8 | 0.019 | 0.87 | 1.16 | 0.86 | 1.17 | 0.77 | 1.32 |
| 43 | 1,3,5-Trimethylbenzene | 56.7 | 0.007 | 0.93 | 1.08 | 0.90 | 1.13 | 0.73 | 1.37 |
| 44 | styrene | 58 | 0.010 | 0.91 | 1.11 | 0.90 | 1.13 | 0.98 | 1.02 |
| 45 | 2-methyl-1-pentene | 63 | 0.002 | 0.81 | 1.25 | 0.70 | 1.49 | 0.81 | 1.28 |
| 46 | trans-2-Butene | 63.2 | 0.014 | 0.84 | 1.22 | 0.82 | 1.25 | 0.76 | 1.35 |
| 47 | cis-2-Pentene | 65 | 0.006 | 0.86 | 1.19 | 0.74 | 1.42 | 0.83 | 1.24 |
| 48 | 1,3-Butadiene | 65.9 | 0.024 | 0.88 | 1.16 | 0.82 | 1.26 | 0.87 | 1.18 |

| 49 | trans-2-Pentene | 67 | 0.009 | 0.88 | 1.16 | 0.63 | 1.63 | 0.75 | 1.38 |
| 50 | β-pinene | 73.5 | 0.010 | 0.90 | 1.12 | 0.81 | 1.26 | 0.92 | 1.10 |
| 51 | isoprene | 99.6 | 0.009 | 0.73 | 1.40 | 0.67 | 1.50 | **0.55** | **1.57** |

\* Unit: $10^{-12}$ cm$^3$ mole$^{-1}$ s$^{-1}$. $k_{OH}$ values were under conditions of 300K. (Carter 2010)

\*\* Unit: ppb. The relative standard derivations (RSDs) were within 10% for the target compounds in all six replicates.

\*\*\* All species were selected for sensitivity tests of initial VOCs to [OH]×t. The reaction rates of these species with OH covered the range of 51 VOCs and were characterized by low, medium and high $k_{OH}$ levels. The sensitivity test results showed that the uncertainty in the estimation of initial VOCs caused by the [OH]×t estimation uncertainty ranged from 0.55 to 1.57.

In Text S2, we have added more details about initial VOC concentration calculations and data reliability.

**"Text S2. Calculation of initial VOCs concentrations**

Photochemical initial concentration (PIC) proposed by Shao et al. (2011), which is calculated based on the photochemical-age approach and has been applied to evaluate the effect of photochemical processing on measured VOC levels. Equation S1 essentially describes the integrated OH exposure (Shao et al., 2011).

$$\int c_{OH} dt = \frac{1}{k_{A,OH} - k_{B,OH}} \left[ \ln\left(\frac{VOC_A}{VOC_B}\right)_{initial} - \ln\left(\frac{VOC_A}{VOC_B}\right) \right] \tag{S1}$$

The initial concentration of species $i$ can be calculated using Equation S2.

$$VOC_{i,\ initial} = \frac{VOC_i}{\exp(-k_{i,OH}) \exp(\int c_{OH} dt)} \tag{S2}$$

Substituting equation 1 into equation 2, then we can get equation S3.

$$VOC_{i,\ initial} = \frac{VOC_i}{\exp(-k_{i,\ OH}) \exp\left(\frac{1}{k_{A,OH} - k_{B,OH}} \left[ \ln\left(\frac{VOC_A}{VOC_B}\right)_{initial} - \ln\left(\frac{VOC_A}{VOC_B}\right) \right]\right)} \tag{S3}$$

Where $C_{OH}$ represents the ambient OH concentration; $k_{A,OH}$ and $k_{B,OH}$ represent the reaction rate of compound A and B with OH radical, respectively; t represents the reaction time of species $i$ in the ambient.

In previous work (Shao et al., 2011; Zhan et al., 2021), the selection of ethylbenzene and xylene as tracers was justified for calculating ambient OH exposure under the following conditions: 1) the concentrations of xylene and ethylbenzene were well correlated (Figure S9), which indicated that they were simultaneously emitted; 2) they had different degradation rates in the atmosphere; and 3) the calculated PICs were in good agreement with those calculated using other tracers (Shao et al., 2011; Zhan et

al., 2021).

In this study, the ethylbenzene/xylene pair was used to calculate ambient OH exposure. As shown in Figure S9, the concentrations of xylene and ethylbenzene are well correlated, which indicates that they are simultaneously emitted. In addition, we compared the PICs according to xylene/ethylbenzene with that using toluene/benzene (Figure S10). The calculated PICs ratio (PIC $_{\text{Xylene/Ethylbenzene}}$ / PIC $_{\text{Toluene/Benzene}}$) varied from 0.5 to 1.5 with a mean value of 0.96. This means the calculated initial VOCs was in good agreement when using different tracers. The mean ratio (0.52, from 0.45 to 0.66) of ethylbenzene/xylene before sunrise was taken as the initial ratio of ethylbenzene/xylene. Sensitivity tests showed that the uncertainty of PICs caused by the OH exposure (from −10% to +10%) ranged from 0.55 to 1.57 (Table S4). Variations of air mass may also affect the VOC ratio. Figure S11 A-D shows the mean concentration distribution of ethylbenzene and xylene in the early morning and the whole day based on potential source contribution function (PSCF) analysis. Xylene showed similar patterns to ethylbenzene in different air mass trajectories and different periods. These results indicate that the emissions of xylene and ethylbenzene were constant throughout the day and variations of air mass should have little influence on the initial ratio of VOCs. The hourly concentrations of ethylbenzene and xylene were used to calculate the concentration of initial VOCs. The initial VOC was calculated by adding the measured VOC concentration and the calculated photochemical loss. Figure S12 shows the diurnal variations of the observed and initial VOCs concentrations from 2014 to 2016. Photochemical loss of VOC occurred mainly during the daytime.

It should be noted that the lifetimes ($1/k_2 c_{\text{OH}}$) of highly reactive VOCs, such as isoprene, greatly depend on the OH exposure. The photochemical ages of isoprene were 0.01–6.21 h (1.26 ± 1.12 h). This value is comparable with previously reported photochemical ages (Shao et al., 2011; Gao et al., 2018). However, the initial concentrations of highly reactive VOCs may be overestimated due to their short lifetimes and should be taken as the upper limits. On the other hand, isoprene is a biogenic VOC, while xylene and ethylbenzene are anthropogenic VOCs. If they do not share the same air mass histories, an additional uncertainty is inevitable for the PICs of

isoprene. As shown in Figure S11, isoprene showed similar patterns to that of xylene and ethylbenzene, which means VOC emissions are evenly distributed in Beijing during our observations. This can be ascribed to the fact that our observation site is a typical urban station. Although isoprene and xylene/ethylbenzene different sources, both them are non-point sources on a city scale. Therefore, the photochemical clock calculated using xylene and ethylbenzene is able to correct the photochemical loss of biogenic VOCs to some extent. It should be noted that uncertainty is inevitable when we estimating the photochemical age (Parrish et al., 2007). However, the aim of this work is to test whether the ML-model can reflect the influence of photochemical loss of VOCs species on $O_3$ modelling. The PICs should provide additional information for understanding $O_3$ formation in the atmosphere.".

**Q2:** Line 50-75: It is easier to read if you divide the paragraph appropriately, e.g., Line50, Line53, Line59, Line64, Line75.

**Reply**: Thank you. To make it is readable, we have started a new paragraph in line 54 before "The observed indicators can be utilized to quickly diagnose the sensitivity regime…" and in line 76 before "Compared to traditional methods, machine learning (ML) is able to capture the" in the revised manuscript.

**Q3:** Line 118-129 and S1: As the composition of VOC species varies greatly from year to year as shown in Fig. 1(F), the analytical reliability is important. Information about VOC measurement in this article is insufficient to understand the reliability. Further information such as observation period and the reason, sampling time or cycle, calibration using standard gas, and method of quality control should be described.

**Reply**: Thank you so much for your good suggestion. To clarify the analytical radiality, we have added more information about VOC measurement, including sampling time, data quality control, calibrations using standard gas. The details are shown in Text S1 in the SI.

"**Text S1. Field observations**

VOCs were measured in May and June from 2014 to 2016 by an online

commercial instrument (GC-866, Chromatotec, France), which consisted of two independent analyzers for $C_2$-$C_6$ and $C_6$-$C_{12}$ hydrocarbons. Both analyzers were equipped with a preconcentration system, a chromatographic column, and a flame ionization detector. The analyzers are located in an air-conditioned room and the sample tubes are wrapped with a heating jackets and insulation to ensure that the temperature remains stable between 22 and 27°C.

The samples were injected into the low carbon ($C_2$-$C_6$) analyzer and the high carbon ($C_6$-$C_{12}$) analyzer, respectively. Isoprene was detected in the components of $C_2$-$C_6$, while α-pinene and β-pinene were detected along with other VOCs of $C_6$-$C_{12}$. In the low carbon analyzer, the samples were adsorbed by a capture tube at -8 °C. The capture tube was then rapidly heated to 220 °C. The samples were introduced into a chromatographic column (id=0.53mm, length=25m) with hydrogen as the carrier gas and detected by a flame ionization detector (FID) detector. In the high carbon analyzer, the samples were adsorbed by a capture tube at room temperature; then the capture tube was heated to 380 °C, introduced into a chromatographic column (id=0.28mm, length=30m) with hydrogen as the carrier gas and finally detected by the same FID detector. The material in the column was $Al_2O_3/Na_2SO_4$.

The signals of VOCs were converted into chromatograms for qualitative and quantitative analysis. Before quantitative analysis, the retention time of each component was carefully checked using the chromatographic analysis software. The instruments were calibrated using both internal and external calibrations. Internally calibration was carried out twice every 24 hours using n-butane, n-hexane and benzene at different flow rates. External calibration was performed monthly using standard gas mixtures of volatile organic compounds (PAMS and TO-14, Linde gas, USA). The concentrations of each species were calculated according to the corresponding working curves with six concentration levels. In this study, total of 51 VOCs (including 21 alkanes, 13 alkenes, 1 alkyne and 16 aromatics) were analyzed within a limit of quantification of 0.002-0.05 ppbv as shown in Table S4. The relative standard derivations (RSDs) were within 10% for each compound among seven replicates.".

**Q4:** Line 129 and S2: Explanation about PIC is insufficient in this article although PIC is important for the results. To calculate PIC, the initial ratio of Ethylbenzene and xylene must be constant. However, they may be emitted from several sources, e.g. painting, mobile exhaust, etc. Please explain why you can use these compounds in this study. And please describe the VOC sampling time which is also important to calculate PIC, because chemical reactions in the air are different in daytime and nighttime.

**Reply**: Thank you very much for your suggestions. We added explanations about PIC calculations in Text S2. Meanwhile, the xylene/ethylbenzene was selected as tracer for the following reasons: 1) the concentrations of xylene and ethylbenzene are well correlated, which indicates that they are simultaneously emitted; 2) they have different degradation rates in the atmosphere; 3) the calculated initial VOCs are in good agreement with those calculated using other tracers, such as toluene/benzene. As shown in Figure R1 or Figure S9, the concentrations of xylene and ethylbenzene correlated well during our observations in this work. In addition, we compared the ratio of the initial concentrations calculated according to the ratio of xylene/ethylbenzene with that using the ratio of toluene/benzene (Figure R2 or S9). Except for several compounds, the ratio of the PICs for most of these VOCs varied within 1.0±0.1. This means the calculated photochemical initial concentrations (PICs) are in good agreement when using different tracers. Sensitivity tests showed that the uncertainty caused by the OH exposure (from −10% to +10%) ranged from 0.55 to 1.57 (Table R1 or Table S4). Figure R3 or Figure S12 shows the calculated diurnal curves of the PICs from 2014 to 2016. Photochemical losses of VOCs occurred prominently during the daytime.

Potential source contribution function (PSCF) analysis has been further carried out to evaluated the possible influence of air mass on the emission ratio of ethylbenzene and xylene. As shown in Figure R4A-D or Figure S11, xylene showed similar pattern to ethylbenzene in the early morning or in the whole day. These results indicate that variations of air mass should have little influence on their initial ratio. In addition, isoprene showed similar patterns to that of xylene and ethylbenzene (Figure R4G-H), which means VOC emissions are evenly distributed in Beijing. This can be ascribed to the fact that our observation site is a typical urban station. Although isoprene and

xylene/ethylbenzene are from biogenic sources and anthropogenic sources, both them are non-point sources on a city scale.

[Figure]

Figure R4. The potential source contribution function (PSCF) maps for ethylbenzene (A and B), xylene (C and D), ratio of xylene to ethylbenzene (E and F), and isoprene

(G and H) arriving in the observation site. The figures A, C, E and G are the results for the morning (05:00 and 06:00), and the figures of B, D, F and H are the results of the whole day (00:00-23:00).

Figure R3 or Figure S12 shows the diurnal variations of the observed and initial VOCs concentrations from 2014 to 2016. From Figure R3, it can be seen that photochemical loss of VOCs occurred prominently during the daytime. We have added information about the sampling time ("VOCs were measured in May and June from 2014 to 2016 by an online commercial instrument…") in **Text S1**.

**Q5:** Figure 1: What are the red lines in (A)?

**Reply**: Thank you. The red lines (arrows) indicate the $O_3$ concentration exceed 74.6 ppbv according to the national ambient air quality standard. We have added an explanation in the caption of Figure 1 "**Figure 1.** Time series of air pollutants and meteorological parameters during observations in Beijing (In A, the red arrows represent the $O_3$ concentration exceed 74.6 ppbv according to the national ambient air quality standard)".

**Q6:** Figure 2: It is difficult to find the difference in (A), (B), and (C). Something will be needed to make it clear.

**Reply**: Thank you. We revised these figures (bolded the lines and increased the color contrast) to make it clearer. Actually, Figure 2(D)-(F) can show their difference for the correlations between modeled and observed $O_3$ concentrations.

**Q7:** Line 203-205: It is unclear that this is what part about Figure 3A.

**Reply**: Thank you. We have adjusted the position of the label in Figure 3 (Figure R5) and the left part is Figure 3A.

[Figure]

**Figure R5.** Percentage of RI for O₃ precursors and meteorological parameters (A) and the top 10 factors with high values of RI in 2014-2016 (B-D: using initial concentrations of VOC species).

**Q8:** Figure 3A: Please explain why RI does not change so much even though the composition of VOCs differs greatly between 2015 and 2016.

**Reply**: Thank you for your good comments. VOCs contributed 64.0%, 58.9%, and 63.3% to the RI, respectively, in 2014, 2015, and 2016. From Figure R5 or Figure 3, it did not change so much among the three years although VOCs concentrations changed obviously. We think this should be ascribed to the production of O₃ in Beijing was still in a VOCs-sensitive regime as shown in Figure R6. This is consistent with previous studies based on transport chemical model (Li et al. 2020) which found VOCs were the dominant contributors to O₃ formation in Beijing. This means that the sensitivity of RI to VOCs concentrations might be not so high. We suppose that it should be more prominent when O₃ formation shifts to a NOx-sensitive regime. However, as shown in Figure R5 (Figure 3B-D), the RI of individual VOC species changed obviously among different years.

[Figure]

**Figure R6.** Ozone formation sensitivity curves from 2014-2016. (A, B, C: calculated by the RF model for 2014, 2015, and 2016, respectively. D: calculated by the OBM for 2015.)

**Q9:** Line 287-289: It is unclear why you can describe that the RF model is better than the box model from Figure S5.

**Reply**: Thank you. We have deleted Figure S5 and revised the statement to "We compared the relative error of simulated MDA8 $O_3$ calculated using the RF and OBM model in 2015, as shown in Figure S8. The mean relative error of simulated MDA8 $O_3$ between RF model and Box model was 15.6%. Hence, a combination of the RF model and initial VOCs species can well depict the sensitivity regime of $O_3$ formation, while the calculated RIs correlate well with the OFPs." in lines 319-323.

**References**

Carter, W.P.L. Development of the SAPRC-07 chemical mechanism. Atmos. Environ., 44, 5324-5335, https://doi.org/10.1016/j.atmosenv.2010.01.026, 2010.

Gao, J., Zhang, J., Li, H., Li, L., Xu, L., Zhang, Y., Wang, Z., Wang, X., Zhang, W., Chen, Y., Cheng, X., Zhang, H., Peng, L., Chai, F., Wei, Y. Comparative study of volatile organic compounds in ambient air using observed mixing ratios and initial mixing ratios taking chemical loss into account – A case study in a typical urban area in Beijing. Sci. Total Environ., 628-629, 791-804, https://doi.org/10.1016/j.scitotenv.2018.01.175, 2018.

Parrish, D.D., Stohl, A., Forster, C., Atlas, E.L., Blake, D.R., Goldan, P.D., Kuster, W.C., de Gouw, J.A. Effects of mixing on evolution of hydrocarbon ratios in the troposphere. J. Geophys. Res. Atmos., 112, https://doi.org/10.1029/2006JD007583, 2007.

Li, K., Jacob, D.J., Shen, L., Lu, X., De Smedt, I., Liao, H. Increases in surface ozone pollution in China from 2013 to 2019: anthropogenic and meteorological influences. Atmos. Chem. Phys., 20, 11423-11433, 10.5194/acp-20-11423-2020, 2020.

Shao, M., Bin, W., Sihua, L., Bin, Y., Ming, W. Effects of Beijing Olympics Control Measures on Reducing Reactive Hydrocarbon Species. Environ. Sci. Technol., 45, 514-519, 10.1021/es102357t, 2011.

---

## Author Response (AR2)

**Reviewer 1#**

In this revised manuscript, the authors addressed most of my previous concerns. Specifically, the revised manuscript now includes more details about the random forest model as well as the relevant measures to mitigate overtraining; the authors also discussed in greater detail the initial VOCs. I appreciate the authors' efforts and it is my opinion that the quality of the manuscript is greatly improved. I recommend the manuscript for publication after the following minor comments are addressed, which are intended to further improve the clarity and the flow:

**Response:** Thank you for your good comments and suggestions. We have carefully responded to all of your **point-by-point** comments and issues and have revised the manuscript accordingly. These revisions are described in detail below.

Line 43: … exhibiting

**Response:** Thank you. We have revised it to "exhibiting" in Line 43.

Line 41-53: consider combining this paragraph with the next for a smoother flow.

**Response:** Thank you. We have combined these two paragraphs to make it smoother.

"…Therefore, it is urgent to develop an accurate and highly efficient method for timely assessing the sensitivity regime of $O_3$ production and evaluating the effectiveness of a potential measure on $O_3$ pollution control. The sensitivity of $O_3$ formation can usually be analysed using observed indicators…"

Line 99-101: This sentence touches one common concern that some machine learning models may be less transparent/interpretable compared to other conventional techniques, thus giving the impression that the authors would discuss this concern in greater detail in this paragraph. Yet, the rest of this paragraph drifted away. Then the next paragraph opens with yet another statement on this "black box" concern. Please rewrite these two paragraphs to improve the logic and flow.

**Response:** Thank you. We have rewritten these two paragraphs to improve the logic and flow.

"Although ML is widely used to understand air pollution, many ML studies have

used total VOCs (TVOCs) to simulate $O_3$ formation and rarely considered the effect of VOC species on $O_3$ formation sensitivity (Feng et al., 2019; Liu et al., 2021; Ma et al., 2021a). Thus, they were unable to identify the chemical reactivity of a single species to $O_3$ formation, which may lead to underestimations or even misunderstandings of the role of VOCs in $O_3$ formation because the same concentration of TVOCs with different compositions may lead to different OPEs. In addition, VOCs react with OH radicals during atmospheric transport, which is the most important sink of VOCs (Carlo et al., 2004; Liu et al., 2020b). Makar et al. (1999) reported that the isoprene emissions were underestimated by up to 40% if the OH oxidation is not considered. Other studies indicated that the initial concentrations of VOCs, which account for the photochemical loss of VOCs during transport, were more representative of pollution levels in the sampling area than the observed VOCs (Yuan et al., 2013; Zhan et al., 2021). However, whether the ML model can identify the connection between the reactivity of VOC species and $O_3$ formation sensitivity has not been clarified.

It should be noted that physical interpretability of the results is an important question when ML models are applied in atmospheric studies (Hou et al., 2022). However, explanations of ML results (e.g., RI) are somewhat vague because ML is a "black-box" model from the point view of chemical mechanism (Hou et al., 2022; Taoufik et al., 2022)…."

Line 235-237: This is a healthy start, but the outcome of this 12-fold cross validation is missing. Please include a figure or table, perhaps in the SI, to archive the consistency of the model performance across all 12 folds. Whether splitting the dataset randomly is a good strategy for cross validation remains a subject of debate, but it is key to archive all key details.

**Response:** Thank you. We have added the results of 12-fold cross validation in **Table R1** or **S6** and performed Pearson correlation calculations as shown in **Table R2** or **S7**. It can be seen in Table R2 that there is consistency in the results between the different folds (ranged from 0.95 to 0.98).

**Table R1**. The RI values of 2015 in all 12 folds

| species number | Fold number | | | | | | | | | | | |
|---|---|---|---|---|---|---|---|---|---|---|---|---|
| | 1 | 2 | 3 | 4 | 5 | 6 | 7 | 8 | 9 | 10 | 11 | 12 |
| Ethane | 1.39 | 1.15 | 1.69 | 2.28 | 1.70 | 1.70 | 1.97 | 1.80 | 1.20 | 1.35 | 1.71 | 1.26 |
| ethene | 1.60 | 1.97 | 1.01 | 1.01 | 1.52 | 2.15 | 0.65 | 2.01 | 1.44 | 2.18 | 2.17 | 1.64 |
| Propane | 0.24 | 0.85 | 0.38 | 1.20 | 1.23 | 1.18 | 0.49 | 0.01 | 1.19 | 0.54 | 0.28 | 1.61 |
| propene | 5.70 | 5.95 | 6.30 | 5.88 | 6.06 | 5.14 | 6.59 | 6.18 | 5.94 | 5.99 | 6.10 | 5.28 |
| iso-Butane | 1.45 | 0.96 | 1.58 | 1.55 | 1.67 | 1.22 | 1.77 | 1.97 | 1.26 | 1.80 | 1.64 | 1.56 |
| n-Butane | 0.05 | 1.08 | 0.35 | 0.32 | 0.35 | 0.61 | 0.27 | 0.24 | 0.60 | 0.05 | 0.09 | 0.05 |
| Acetylene | 1.50 | 1.11 | 1.01 | 1.03 | 0.66 | 0.00 | 0.96 | 0.96 | 0.23 | 1.05 | 1.02 | 1.81 |
| trans-2-Butene | 1.30 | 2.26 | 2.37 | 2.09 | 1.51 | 2.22 | 2.23 | 2.30 | 1.93 | 2.42 | 2.60 | 2.34 |
| 1-Butene | 1.71 | 1.29 | 0.89 | 1.13 | 1.20 | 1.34 | 1.31 | 1.17 | 1.70 | 1.20 | 0.93 | 1.42 |
| Cyclopentane | 0.00 | 0.00 | 0.25 | 0.00 | 0.25 | 0.25 | 0.24 | 0.24 | 0.27 | 0.00 | 0.25 | 0.24 |
| cis-2-Butene | 0.86 | 1.05 | 1.33 | 1.09 | 1.43 | 1.65 | 1.65 | 1.23 | 1.60 | 0.81 | 1.44 | 1.15 |
| iso-Pentane | 0.00 | 0.00 | 0.30 | 0.00 | 0.25 | 0.25 | 0.25 | 0.24 | 0.23 | 0.00 | 0.23 | 0.27 |
| 1,3-Butadiene | 0.58 | 1.31 | 0.86 | 1.10 | 0.85 | 1.17 | 0.69 | 0.99 | 1.36 | 0.91 | 1.08 | 0.93 |
| trans-2-Pentene | 0.99 | 0.75 | 1.26 | 0.78 | 1.16 | 1.18 | 1.03 | 1.01 | 0.69 | 1.17 | 1.12 | 1.23 |
| 1-Pentene | 0.28 | 0.70 | 0.81 | 0.77 | 0.59 | 1.06 | 0.61 | 0.92 | 0.86 | 0.56 | 1.07 | 0.49 |
| cis-2-Pentene | 1.29 | 1.37 | 1.16 | 1.21 | 1.26 | 0.84 | 1.23 | 1.76 | 1.55 | 1.09 | 1.25 | 0.98 |
| 2,3-Dimethylbutane | 0.00 | 0.00 | 0.00 | 0.00 | 0.00 | 0.00 | 0.00 | 0.24 | 0.00 | 0.26 | 0.00 | 0.00 |
| n-hexane | 1.27 | 1.42 | 1.69 | 1.46 | 1.42 | 1.47 | 1.30 | 1.62 | 0.82 | 2.00 | 1.08 | 0.91 |
| isoprene | 5.30 | 4.16 | 4.96 | 5.25 | 4.98 | 5.39 | 5.98 | 5.25 | 5.44 | 5.06 | 5.34 | 6.24 |
| 2-methyl-1-pentene | 2.27 | 2.51 | 2.08 | 1.93 | 2.36 | 2.09 | 2.10 | 1.74 | 2.35 | 2.28 | 2.26 | 2.17 |
| 2,2-Dimethylbutane | 0.78 | 0.63 | 0.65 | 0.68 | 0.87 | 0.63 | 0.41 | 0.57 | 0.81 | 0.71 | 0.63 | 0.60 |
| Benzene | 0.35 | 0.79 | 0.96 | 1.37 | 0.84 | 1.12 | 0.86 | 0.83 | 0.80 | 1.10 | 1.10 | 0.79 |
| cyclohexane | 1.71 | 1.40 | 1.18 | 1.20 | 1.46 | 1.06 | 1.55 | 1.09 | 0.86 | 1.21 | 1.22 | 1.61 |

| | | | | | | | | | | | |
|---|---|---|---|---|---|---|---|---|---|---|---|
| 2,3-Dimethylpentane | 2.26 | 1.61 | 2.39 | 1.65 | 2.23 | 1.93 | 1.68 | 1.56 | 1.89 | 2.16 | 2.01 | 2.21 |
| 3-Methylhexane | 1.05 | 0.36 | 0.63 | 0.98 | 0.48 | 0.92 | 0.84 | 0.69 | 0.54 | 0.45 | 0.90 | 0.66 |
| 2,2,4-Trimethylpentane | 1.59 | 1.64 | 1.62 | 1.72 | 1.28 | 1.33 | 1.46 | 1.59 | 1.85 | 2.07 | 1.84 | 1.71 |
| n-heptane | 1.19 | 1.13 | 1.15 | 1.16 | 1.36 | 0.97 | 0.95 | 0.60 | 1.06 | 1.27 | 1.17 | 0.83 |
| methylcyclohexane | 0.34 | 0.56 | 0.97 | 0.69 | 0.50 | 0.55 | 0.75 | 0.42 | 0.57 | 0.70 | 0.81 | 0.86 |
| 2,3,4-trimethylpentane | 0.76 | 0.76 | 0.87 | 1.11 | 0.89 | 0.90 | 0.81 | 0.67 | 0.95 | 0.81 | 0.88 | 0.97 |
| Toluene | 0.79 | 0.47 | 0.63 | 0.50 | 0.75 | 0.39 | 0.49 | 0.64 | 0.91 | 0.71 | 0.68 | 0.52 |
| 2-methylhexane | 1.56 | 1.69 | 1.39 | 1.32 | 1.61 | 1.73 | 1.37 | 1.50 | 1.02 | 1.93 | 1.08 | 1.25 |
| 3-methylhexane | 0.47 | 0.31 | 0.51 | 0.83 | 0.79 | 0.70 | 0.54 | 0.78 | 0.47 | 0.50 | 0.67 | 0.61 |
| n-octane | 0.00 | 0.00 | 0.00 | 0.00 | 0.00 | 0.00 | 0.00 | 0.00 | 0.00 | 0.00 | 0.00 | 0.00 |
| ethylbenzene | 0.73 | 0.81 | 1.10 | 0.83 | 0.48 | 0.72 | 0.51 | 0.61 | 0.68 | 0.79 | 0.79 | 0.92 |
| m,p-Xylene | 0.52 | 0.31 | 0.83 | 0.22 | 0.32 | 0.24 | 0.45 | 0.36 | 0.01 | 0.53 | 0.57 | 0.22 |
| styrene | 0.31 | 0.41 | 0.23 | 0.27 | 0.25 | 0.26 | 0.33 | 0.33 | 0.37 | 0.17 | 0.29 | 0.28 |
| o-xylene | 0.59 | 0.83 | 0.88 | 0.61 | 0.62 | 0.98 | 0.63 | 0.64 | 0.86 | 0.83 | 0.66 | 0.30 |
| n-nonane | 1.19 | 1.30 | 1.38 | 1.68 | 1.34 | 1.57 | 1.30 | 1.14 | 1.48 | 1.27 | 1.27 | 1.73 |
| iso-Propylbenzene | 0.33 | 0.13 | 0.49 | 0.32 | 0.51 | 0.25 | 0.29 | 0.11 | 0.01 | 0.34 | 0.39 | 0.22 |
| a-pinene | 2.10 | 1.47 | 1.67 | 2.02 | 1.87 | 1.97 | 1.77 | 2.43 | 1.53 | 1.72 | 1.60 | 1.83 |
| n-Propylbenzene | 2.03 | 1.91 | 1.88 | 2.15 | 1.89 | 2.24 | 1.69 | 1.86 | 1.69 | 1.88 | 1.91 | 2.23 |
| m-ethyl toluene | 1.54 | 0.95 | 0.91 | 1.08 | 0.79 | 0.90 | 0.64 | 1.02 | 0.99 | 0.97 | 0.64 | 0.95 |
| p-ethyltoluene | 0.55 | 0.64 | 1.22 | 0.74 | 1.06 | 0.98 | 0.74 | 1.01 | 0.73 | 0.84 | 0.59 | 0.53 |
| 1,3,5-Trimethylbenzene | 0.80 | 1.30 | 0.80 | 1.06 | 0.78 | 1.01 | 1.35 | 0.97 | 1.18 | 1.18 | 0.86 | 1.00 |
| o-ethyl toluene | 1.00 | 0.98 | 0.98 | 0.73 | 0.71 | 0.91 | 1.09 | 0.59 | 0.84 | 0.60 | 0.88 | 1.03 |
| β-pinene | 0.20 | 0.06 | 0.23 | 0.01 | 0.25 | 0.14 | 0.02 | 0.46 | 0.05 | 0.04 | 0.25 | 0.01 |
| 1,2,4-trimethyl benzene | 1.38 | 1.67 | 1.54 | 1.65 | 2.69 | 1.34 | 1.69 | 1.56 | 1.03 | 1.43 | 1.60 | 0.91 |
| n-decane | 1.60 | 1.84 | 2.02 | 0.93 | 1.52 | 1.88 | 1.91 | 1.58 | 1.97 | 1.53 | 1.93 | 1.58 |
| 1,2,3-trimethyl benzene | 1.14 | 1.05 | 1.80 | 1.22 | 0.96 | 1.70 | 1.37 | 1.11 | 0.67 | 0.88 | 1.53 | 1.32 |

| | | | | | | | | | | | | |
|---|---|---|---|---|---|---|---|---|---|---|---|---|
| m-diethyl benzene | 0.26 | 0.00 | 0.25 | 0.26 | 0.24 | 0.36 | 0.00 | 0.34 | 0.00 | 0.26 | 0.00 | 0.00 |
| p-diethyl benzene | 1.19 | 0.95 | 0.95 | 0.69 | 0.92 | 0.80 | 0.79 | 0.92 | 0.80 | 0.97 | 0.75 | 0.86 |
| $NO_x$ | 16.35 | 16.32 | 14.22 | 15.74 | 15.93 | 11.94 | 14.37 | 14.00 | 15.79 | 14.78 | 13.75 | 14.23 |
| T | 10.25 | 9.69 | 9.45 | 9.69 | 9.18 | 10.77 | 10.61 | 10.74 | 9.81 | 8.76 | 8.71 | 9.50 |
| RH | 3.93 | 4.27 | 3.36 | 4.31 | 3.85 | 3.68 | 3.63 | 3.62 | 3.96 | 3.61 | 3.11 | 4.10 |
| SR | 3.35 | 3.64 | 3.94 | 3.86 | 3.34 | 3.73 | 3.77 | 3.72 | 3.44 | 3.30 | 4.25 | 3.83 |
| WS&WD | 4.28 | 3.08 | 4.29 | 3.15 | 2.63 | 3.59 | 3.42 | 4.05 | 4.13 | 3.52 | 4.63 | 3.71 |
| $PM_{2.5}$ | 1.33 | 1.23 | 1.00 | 1.24 | 2.46 | 1.78 | 1.81 | 1.03 | 2.20 | 2.99 | 1.79 | 2.00 |
| CO | 3.00 | 4.03 | 2.97 | 2.72 | 2.91 | 3.88 | 3.78 | 3.64 | 3.41 | 3.05 | 3.54 | 2.49 |

62 **Note:** To verify the consistency between the different folds, we calculated the Pearson correlation coefficient ($r$) between the different folds as
63 shown in **Table R2**.

64

65 **Table R2.** The Pearson correlation coefficient ($r$) between different folds in 2015

| Fold number | 1 | 2 | 3 | 4 | 5 | 6 | 7 | 8 | 9 | 10 | 11 | 12 |
|---|---|---|---|---|---|---|---|---|---|---|---|---|
| 1 | 1.00 | 0.98 | 0.98 | 0.98 | 0.98 | 0.95 | 0.98 | 0.98 | 0.98 | 0.98 | 0.97 | 0.98 |
| 2 | 0.98 | 1.00 | 0.98 | 0.98 | 0.97 | 0.95 | 0.97 | 0.97 | 0.98 | 0.98 | 0.96 | 0.96 |
| 3 | 0.98 | 0.98 | 1.00 | 0.98 | 0.97 | 0.96 | 0.98 | 0.98 | 0.98 | 0.98 | 0.98 | 0.97 |
| 4 | 0.98 | 0.98 | 0.98 | 1.00 | 0.98 | 0.96 | 0.98 | 0.98 | 0.98 | 0.97 | 0.97 | 0.98 |
| 5 | 0.98 | 0.97 | 0.97 | 0.98 | 1.00 | 0.96 | 0.98 | 0.97 | 0.98 | 0.98 | 0.97 | 0.97 |
| 6 | 0.95 | 0.95 | 0.96 | 0.96 | 0.96 | 1.00 | 0.98 | 0.98 | 0.96 | 0.95 | 0.97 | 0.96 |
| 7 | 0.98 | 0.97 | 0.98 | 0.98 | 0.98 | 0.98 | 1.00 | 0.98 | 0.98 | 0.97 | 0.98 | 0.98 |
| 8 | 0.98 | 0.97 | 0.98 | 0.98 | 0.97 | 0.98 | 0.98 | 1.00 | 0.98 | 0.97 | 0.98 | 0.97 |
| 9 | 0.98 | 0.98 | 0.98 | 0.98 | 0.98 | 0.96 | 0.98 | 0.98 | 1.00 | 0.98 | 0.97 | 0.98 |
| 10 | 0.98 | 0.98 | 0.98 | 0.97 | 0.98 | 0.95 | 0.97 | 0.97 | 0.98 | 1.00 | 0.97 | 0.97 |
| 11 | 0.97 | 0.96 | 0.98 | 0.97 | 0.97 | 0.97 | 0.98 | 0.98 | 0.97 | 0.97 | 1.00 | 0.97 |
| 12 | 0.98 | 0.96 | 0.97 | 0.98 | 0.97 | 0.96 | 0.98 | 0.97 | 0.98 | 0.97 | 0.97 | 1.00 |

66

Line 286-291: This is interesting. Is the decrease in RH usually accompanied with changes in other parameters/conditions? Perhaps a change in weather system, cloud cover (i.e. change in radiation), etc? Keep in mind that the features are not always independent variables (certainly not the case in this work, which is perfectly fine). Say, if two features A and B are equally important, the algorithm may give high importance to A (or B) but very low importance to B (or A), thus giving the wrong impression that A (or B) is important but B (or A) is not. I would be a little surprised if the negative response of ozone to RH is really driven by $NO_2$ uptake under high RH, after all $NO_2$ is only moderately soluble.

**Response:** Thank you for your good comments. We agree with you that the features are not always independent variables. RH may change accompanied by other parameters/conditions. We correlated RH with solar radiation (SR) as you suggested. However, RH and SR showed a weak correlation ($r < 0.1$). Meanwhile, we tested the independence of RH and SR as shown in Table R3 or S4, which showed that RH an SR are independent of each other. As shown in Table R2, the algorithm in this work is pretty stable, which indirectly suggests the independence of RH on other parameters, unless great variations of the RI of RH should be observed. In addition, Hu et al. (2011) found that RH was negatively related to the rate constant of HONO formation. Thus, RH also affects $O_3$ formation by influencing atmospheric OH radicals from HONO photolysis. Therefore, the negative response of $O_3$ to RH was not just driven by $NO_2$ uptake under high RH, but also by the deposition of $O_3$ and the decrease of HONO formation rate constant. We added the discussion in Lines 289-296.

"In addition, it has been shown that RH is negatively related to the rate constant of HONO formation (Hu et al., 2011). Thus, RH might also affect the $O_3$ formation by influencing atmospheric OH radicals from photolysis of HONO. It should be noted that the negative response of ozone to RH might also be resulted from the dependence of RH on other parameters/conditions, such as SR. However, RH and SR showed a bad correlation ($r < 0.1$). We further tested the dependence of the RI on RH and SR with or without the counterpart as input. The stable RI values (Table S4) mean that RH and SR are independent from each other."

**Table R3.** Independence test between RH and SR

| name | RI value | | |
|---|---|---|---|
| | RH and SR as input | RH as input | SR as input |
| RH | 0.68 | 0.68 | / |
| SR | 0.76 | / | 0.76 |

Code and data availability: The authors mentioned in the response that the random forest model used in this work is not based on widely used software packages/platforms (such as python/scikit-learn) but is developed in-house in MATLAB (this is impressive, if implemented properly). The vast majority of the dataset used in work is also not publicly accessible as of now. Please refer to the journal policy/guidelines on code and data availability. Given the absolute critical role the random forest model is playing in this work, I strongly recommend that the authors deposit the random forest model code in FAIR (Findable, Accessible, Interoperable, and Reusable)-aligned reliable public repositories.

**Response:** Thank you. We have deposited the random forest model code on GitHub (https://github.com/z-12/amt-2021-367.git). We have revised it to "The code can be seen in GitHub (https://github.com/z-12/amt-2021-367.git)." in Lines 354-355.

**References**

Hu, G., Xu, Y., Jia, L. Effects of relative humidity on the characterization of a photochemical smog chamber. J. Environ. Sci., 23, 2013-2018, https://doi.org/10.1016/S1001-0742(10)60665-1, 2011.

---

## Author Response (AR3)

Comments to the author:

Thank you for amending the manuscript and uploading code to Github. It appears that some of the functions you uploaded are stock functions from Matlab's machine learning toolbox. I suggest you make the following changes:

1) Please mention explicitly in Sect. 2.2 that you are using the Random forest methods in Matlab's Statistics and Machine Learning toolbox.

**Response:** Thank you. We have added a sentence "In this work, we performed $O_3$ and RI calculations using the RF method in MATLAB's Statistics and machine learning toolbox." in lines 145-147 in Section 2.2

2) I recommend removing TreeBagger and CompactRegressionTree from your github archive, as those functions are copyrighted by Matlab.

**Response:** Thank you. We have removed the TreeBagger and CompactRegressionTree files from our github archive.

[Figure]

Figure R1. Screenshot of the code storage page.

3) Add a few comments to the top of your random forest script, including author,

19   creation date, a note about needing the toolbox to use this code, and maybe a URL or

20   DOI for your paper. This will help others who might want to use your code, thereby

21   increasing the impact of your work.

22   **Response:** Thank you. We have added author name, creation data, notes and a URL in

23   the top of your random forest script.

```
1   # Creation date: 2022.02.09
2   # Author: Junlei Zhan
3   # Manuscript: https://doi.org/10.5194/amt-2021-367
4   # Note: This code consists of six parts:
5   #         data import, cross-validation partitioning, data normalisation, data
6   #         training, data denormalisation and data output. We have annotated the
7   #         corresponding parts of the code.
```

24

25              Figure R2. Screenshot of the top of the random forest file.

26